# STEERING PROTEIN FAMILY DESIGN THROUGH PROFILE BAYESIAN FLOW

**Jingjing Gong**[1*]    **Yu Pei**[1*]    **Siyu Long**[1*]    **Yuxuan Song**[1*]    **Zhe Zhang**[1]
**Wenhao Huang**[1]    **Ziyao Cao**[1]    **Shuyi Zhang**[2]    **Hao Zhou**[1]    **Wei-Ying Ma**[1]
[1] Institute of AI Industry Research (AIR), Tsinghua University
[2] School of Pharmaceutical Sciences, Tsinghua University
{jjgongjj,yupei.wp,yxsong0816,longlonglongguy}@gmail.com
{zhouhao,maweiying}@air.tsinghua.edu

## ABSTRACT

Protein family design emerges as a promising alternative by combining the advantages of de novo protein design and mutation-based directed evolution. In this paper, we propose ProfileBFN, the Profile Bayesian Flow Networks, for specifically generative modeling of protein families. ProfileBFN extends the discrete Bayesian Flow Network from an MSA profile perspective, which can be trained on single protein sequences by regarding it as a degenerate profile, thereby achieving efficient protein family design by avoiding large-scale MSA data construction and training. Empirical results show that ProfileBFN has a profound understanding of proteins. When generating diverse and novel family proteins, it can accurately capture the structural characteristics of the family. The enzyme produced by this method is more likely than the previous approach to have the corresponding function, offering better odds of generating diverse proteins with the desired functionality.

## 1 INTRODUCTION

Protein design stands as a crucial problem with far-reaching implications. In particular, it holds the potential to significantly accelerate progress in numerous areas such as precision medicine and synthetic biology (Kosorok & Laber, 2019; Johnson et al., 2021; Benner & Sismour, 2005). Recently, artificial intelligence (AI) has brought new possibilities and breakthroughs to protein design (Jumper et al., 2021; Abramson et al., 2024; Lin et al., 2023; Hayes et al., 2024). AI-powered techniques are increasingly being employed to accelerate the process and enhance the accuracy of protein design. The ability to design proteins with specific functions using AI is not only a scientific pursuit but also a practical necessity for addressing various challenges in these fields.

Protein design often involves a combination of de novo design and mutation-based directed evolution. De novo design generates proteins almost from scratch, offering novel protein sequences that expand the diversity of protein libraries (Watson et al., 2023; Dahiyat & Mayo, 1997). Although it may have a lower success rate in wet lab experiments, it is valuable for creating starting points that can be further optimized. Directed evolution (Arnold, 1998; Packer & Liu, 2015) is effective in developing proteins with enhanced functions in vitro. However, the scope of exploration within the vast protein sequence space remains limited due to constraints in both the throughput of library creation and the subsequent screening or selection processes (Wang et al., 2021; Bloom & Arnold, 2009).

In this context, protein family design emerges as an approach that combines the strengths of both methods. By generating protein candidates based on multiple existing functional proteins, it explores protein space more broadly than mutation-based methods alone while utilizing established functional information. This generative process allows for the creation of diverse libraries without being limited to sequences closely related to a single wild type. Similar methods, such as ProtMamba (Sgarbossa et al., 2024), PoET (Truong Jr & Bepler, 2023) and EvoDiff (Alamdari et al.,

---

*Equal Contribution. Correspondence to Hao Zhou (zhouhao@air.tsinghua.edu).

2023), also aim to balance innovation with reliability in protein design. Overall, protein family design fits within the library creation and optimization pipeline, providing a powerful tool for generating diverse protein candidates that can be further refined through directed evolution.

Recently, single protein sequence modeling has dominated the area due to the analogy to the task of the language model. Hence, there is also rising interest in transferring the techniques from language modeling to protein modeling (Truong Jr & Bepler, 2023; Madani et al., 2023; 2020; Nijkamp et al., 2023; Jumper et al., 2021). In contrast, we believe that directly applying the natural language modeling paradigm could be sub-optimal for the protein sequence distribution with very complex global spatial correlation and constraint. In this paper, we consider integrating the evolutionary information from the MSA[1] (Multiple Sequence Alignment) motivated by previous literature (Rao et al., 2021; Alamdari et al., 2023). However, MSA lies in a specific data type, *i.e.* a set of sequences, and could vary and hold large length and depth which could bring in practical barriers for efficiently processing the information with a scaled model.

To address the above concern and bring a fresh perspective to the protein family generative modeling, we propose the Profile Bayesian Flow Networks (ProfileBFN), which achieves effective yet efficient Protein Family Design by: (i) proposing to use MSA profile (the distribution of MSA) instead of MSA for probabilistic generative modeling, which avoids the heavily direct training of MSA data.[2] (ii) ProfileBFN extends the conventional discrete Bayesian Flow Network (BFN) from an MSA profile perspective. We formally re-derive the new Bayesian flow and loss terms, tailoring it from the perspective of protein family modeling. (iii) ProfileBFN could escape the heavy construction of large-scale MSA data by training on single protein sequences. Thanks to the mathematical nature of the ProfileBFN, we could generalize the one-hot representation of single sequences as a degenerative profile, which enables the ProfileBFN to be flexible for both single sequence and multiple sequence profiles as inputs.

We evaluate ProfileBFN on a multitude of benchmarks and find that ProfileBFN has the following impressive advantages: (i) ProfileBFN ensures structural conservation while providing the most diverse and novel family protein generation results. For characterizing family structural features, sequences generated by ProfileBFN even surpass the MSA search relied upon by AlphaFold2. (ii) In the evaluation of generating functional enzyme proteins, compared to previous advanced methods, ProfileBFN is more likely than the previous approach to have the corresponding function, offering better odds of generating diverse proteins with the desired functionality. (iii) In the aspect of protein representation, ProfileBFN outperforms all PLMs under the same parameter scale, demonstrating its profound understanding of proteins.

## 2 PRELIMINARIES

### 2.1 REPRESENTING PROTEIN FAMILY AS MSA PROFILES

Multiple Sequence Alignments (MSAs) (Edgar & Batzoglou, 2006) are commonly used to capture the evolutionary relationship between protein sequences within a family, it have been widely used in various aspects of protein modeling, including protein sequence analysis (Gromiha, 2010), structure prediction, function prediction, and protein design.

In the context of this paper, a MSA is a set of homologous protein sequences that are aligned to each other. Formally speaking, given a set of $n$ protein sequences, the MSA is a matrix $\boldsymbol{X} \in \{0, \cdots, K\}^{n \times m}$, where $m$ is the length of the aligned protein sequences, and $\boldsymbol{X}_{ij}$ is the $j$-th amino acid in the $i$-th aligned protein sequence.

The MSA profile $\{\boldsymbol{P}^{(i)}\}_{i=1}^{m} \subset \Delta^{K}$, where $\Delta^{K}$ represents the space of k-dimensional simplex, $\boldsymbol{P}$ is calculated as follows:

$$\boldsymbol{P}_k^{(i)} = \frac{1}{n} \sum_{j=1}^{n} \mathbf{1}_{(\boldsymbol{X}_{ji=k})} \tag{1}$$

---

[1]MSA is commonly used to capture the evolutionary relationship between protein sequences within a family.

[2]This is analogous to directly calculating the Schrödinger equation and making estimations using density functional theory.

Where $K$ is the alphabet size of amino acids, $\boldsymbol{P}_k^{(i)}$ is the frequency of amino acid $k$ at position $i$ in the MSA, and $\mathbf{1}_{(\cdot = \cdot)}$ is the Kronecker delta function.

## 2.2 BAYESIAN FLOW NETWORKS

Bayesian Flow Networks (BFNs) (Graves et al., 2023) introduce a new type of generative model from a transmission perspective. In a simpler language, a sender leaks it's information through the noisy process of $z_i \sim q(\cdot|\mathrm{x}; \omega)$. An observer then receives the leaked information and updates its belief about the variable x through Bayesian update and obtain a belief about x: $p(\mathrm{x}|\mathrm{z}_{1:n})$. In the context of a bits-back coding transmission scheme, the total number of nats required to transmit x with $\mathrm{z}_{1:n}$ serving as intermediate latents can be expressed as $-\log p(\mathrm{z}_{1:n}) - \log p(\mathrm{x}|\mathrm{z}_{1:n})$. The process also incorporates $-\log q(\mathrm{z}_{1:n}|\mathrm{x})$ nats returned to the sender, thus yielding the expected marginal nats necessary to transmit data from $p(\mathrm{x})$, which corresponds to the negative Variational Lower Bound (VLB), as:

$$\mathbb{E}_{p(\mathrm{x})}\mathbb{E}_{q(\mathrm{z}_{1:n}|\mathrm{x}; \omega)}\left[-\log p(\mathrm{z}_{1:n}) - \log p(\mathrm{x}|\mathrm{z}_{1:n}) + \log q(\mathrm{z}_{1:n}|\mathrm{x}; \omega)\right]$$
$$= \mathbb{E}_{p(\mathrm{x})}\left[D_{\mathrm{KL}}(q(\mathrm{z}_{1:n}|\mathrm{x}; \omega)||p(\mathrm{z}_{1:n})) - \mathbb{E}_{q(\mathrm{z}_{1:n}|\mathrm{x}; \omega)}\log p(\mathrm{x}|\mathrm{z}_{1:n})\right] = -\texttt{VLB} \quad (2)$$

As $p(\mathrm{z}_{1:n})$ can be decomposed auto-regressively with a neural network $p_\phi$, where $\phi$ is the governing parameter of the neural network, the loss is

$$-\texttt{VLB}(\phi) = \mathbb{E}_{p(\mathrm{x})}\left[\sum_{i=1}^{n} D_{\mathrm{KL}}(q(\mathrm{z}_i|\mathrm{x}; \omega)||p_R(\mathrm{z}_i|\mathrm{z}_{1:i-1}; \phi)) - \mathbb{E}_{q(\mathrm{z}_{1:n}|\mathrm{x}; \omega)}\log p_\phi(\mathrm{x}|\mathrm{z}_{1:n})\right] \quad (3)$$

The $-\texttt{VLB}(\phi)$ is the expected marginal nats required to transfer a data sample from $p(\mathrm{x})$. The loss can be derived into a simpler form:

$$\mathcal{L}(\mathrm{x}) = \frac{1}{2}\beta'(t)K||p_\phi - \boldsymbol{e}_\mathrm{x}||^2 \quad (4)$$

The Bayesian flow required to train the network is:

$$p_F(\boldsymbol{\theta}|\mathrm{x}; t) = \mathbb{E}_{\mathcal{N}(\mathbf{y}|K\beta(t)\boldsymbol{e}_\mathrm{x}, \beta(t)\mathcal{C})}\delta\left(\boldsymbol{\theta} - \frac{e^\mathbf{y}\boldsymbol{\theta}_0}{\sum_{k=1}^{K} e^{\mathbf{y}_k}(\boldsymbol{\theta}_0)_k}\right) \quad (5)$$

Where $\boldsymbol{\theta}$ is the governing parameter of the belief of the variable x. $\mathcal{C}$, is the covariance matrix of the multivariate Gaussian distribution. $\delta(\cdot - \boldsymbol{\theta})$ is a dirac delta function that is zero everywhere except at $\boldsymbol{\theta}$. For detailed easy to understand derivation refer to Appendix A.1.

## 3 METHOD

To generate a protein that belongs to a specific protein family, it's crucial to leverage the information embedded within that family. As introduced in Section 2.1, a profile serves as an effective summary of a protein family's multiple sequence alignment (MSA). Utilizing profiles allows us to harness the collective information of the entire protein family without incurring additional computational costs compared to single-sequence models. However, constructing a training set of MSA profiles is computationally expensive (Liu et al., 2009; Nag & Karforma, 2016).

We introduce our proposed ProfileBFN model, which unifies single-sequence one-hot encoding as a special case of a profile. This innovative approach enables us to train on single protein sequences while sampling with protein family profiles. Consequently, we can bypass the need to construct an MSA profile training set, offering a more efficient and practical solution. Henceforth, we define a profile as a list of PMFs and for simplicity, refer to a PMF as a profile.

### 3.1 THE PROPOSED PROFILEBFN

In the original discrete BFN (Graves et al., 2023), the emitted sample x can be viewed as being drawn from a degenerate profile where each component has all its probability mass concentrated on

a single category. In this work, we extend the discrete BFN to accommodate the input of generalized profiles. This generalization allows for seamless integration with the processing of protein family profiles.

To enable new capabilities, it is necessary to derive a new Bayesian flow and a corresponding loss term. The main intuition behind this is to sample from a generalized profile, pass it through a noisy channel, and then have the parameterized network make predictions based on the received evidence. The Bayesian flow for profile modelling is as below, and the derivation and proof can be found in Appendix A.2.

**Theorem 3.1.** *Given a discrete noisy channel* $q(z_i|\boldsymbol{\rho}; \omega_i) = \frac{1-\omega_i}{K} + \omega_i \boldsymbol{\rho}(z)$ *where* $\boldsymbol{\rho}, \sum_x \boldsymbol{\rho}_x = 1, \forall \boldsymbol{\rho}_x \geq 0$ *is a certain profile, with* $\omega_i^2 = \int_{(i-1)/n}^{i/n} \mu(\tau)^2 d\tau, \beta(t) = \int_0^t \mu^2(\tau) d\tau (1 \geq t \geq 0), \mu(\tau) > 0, \forall \tau$, *and* $\beta(1)$ *bounded, when* $n \to +\infty$, *the continuous time discrete Bayesian flow is:*

$$p_F(\boldsymbol{\theta}|\boldsymbol{\rho};t) = \underset{\mathcal{N}(\mathbf{y}|K\beta(t)\boldsymbol{\rho},\beta(t)\mathcal{C})}{\mathbb{E}} \delta\left(\boldsymbol{\theta} - \frac{e^{\mathbf{y}}\boldsymbol{\theta}_0}{\sum_{k=1}^{K} e^{\mathbf{y}_k}(\boldsymbol{\theta}_0)_k}\right) \tag{6}$$

*Where* $\boldsymbol{\theta}$ *is the accumulated information about the profile* $\boldsymbol{\rho}$. $\mathcal{C} \in \mathbb{R}^{K \times K}, \mathcal{C}_{ij} = K\mathbf{1}_{i=j} - 1$, *is the covariance matrix of the multivariate Gaussian distribution.* $\delta(\cdot - \boldsymbol{\theta})$ *is Dirac delta function that is zero everywhere except at* $\boldsymbol{\theta}$.

Where $\boldsymbol{\rho} \in \Delta^{K-1}$ is a profile which can also be viewed as Probability Mass Function (PMF) with K possible categories, this is the different part compared to vanilla discrete Bayesian flow (Eq. 5).

Additionally, we derive the new loss function as below.

**Theorem 3.2.** *Given a discrete noisy channel* $q(z|\boldsymbol{\rho}) = \frac{1-\omega}{K} + \omega\boldsymbol{\rho}(z), p(z) = \frac{1-\omega}{K} + \omega p_{\phi}(z), \omega > 0$, *where* $\boldsymbol{\rho}, \sum_x \boldsymbol{\rho}_x = 1, \forall \boldsymbol{\rho}_x \geq 0$ *is a certain profile, with* $n\omega^2 = \beta$ *bounded,*

$$\lim_{n \to +\infty} n D_{\mathrm{KL}}(q(z|\boldsymbol{\rho})||p(z)) = \frac{1}{2}\beta K||p_{\phi} - \boldsymbol{\rho}||^2 \tag{7}$$

*For a more general case where* $\omega(t)$ *changes through time, with* $\beta(t) = \int_0^t \omega^2(\tau) d\tau, 1 \geq t \geq 0$, *and* $\beta(1)$ *bounded, the limit of the KL divergence is:*

$$\lim_{n \to +\infty} n D_{\mathrm{KL}}(q(z|\boldsymbol{\rho};t)||p(z;t)) = \frac{1}{2}\beta'(t)K||p_{\phi} - \boldsymbol{\rho}||^2 \tag{8}$$

There is only little change by substituting $e_x$ to $\boldsymbol{\rho}$ with respect to Eq. 4.

From Eq. 15, $p_{\phi} = f_{\phi}(\boldsymbol{\theta}^{(1)}, \cdots, \boldsymbol{\theta}^{(m)})$ represents a neural network, where $\boldsymbol{\theta}^{(i)}$ is the $i$th accumulated information about the profile. The primary purpose of the network is to model the interdependency between independently accumulated information about the profiles.

## 3.2 TRAINING WITH PROFILE AS INPUT

As introduced in Section 2.1 $\{\boldsymbol{P}^{(i)}\}_{i=1}^m \subset \Delta^{K-1}$ is the profile, where $m$ is the length of the protein sequence, and $K$ is the alphabet size of amino acids. $\boldsymbol{P}^{(i)}$ is the probability mass function of the $i$-th position in the MSA profile, indicating the frequency of each amino acid at the $i$-th position in the MSA.

**Unified Profile Representation**   In the special case where the MSA contains only a single sequence, the profile at each position $\boldsymbol{P}^{(i)}$ becomes a one-hot vector. This scenario simplifies to determining the precise amino acid at each position without ambiguity. However, for typical MSAs with multiple sequences, $\boldsymbol{P}^{(i)}$ provides a richer representation reflecting the variability and conservation of amino acids across the alignment.

**ProfileBFN for Protein Generative Modeling**   From Theorem 3.2, it is easy to arrive at the objective function for the training of protein family profile:

$$\mathcal{L}(\boldsymbol{P}) = \sum_{i=1}^{m} \frac{1}{2}\beta'(t)K||\boldsymbol{P}_{\phi}^{(i)} - \boldsymbol{P}^{(i)}||^2 \tag{9}$$

The $\boldsymbol{P}_{\boldsymbol{\phi}}^{(i)}$ is the network part, where it takes independently accumulated information about the profiles $\boldsymbol{\theta}_t^{(i)}$ as input and tries to correlate and guess the true profile. The accumulated information about the profile $\boldsymbol{\theta}_t^{(i)}$ can be computed through the Bayesian flow: $\boldsymbol{\theta}_t^{(i)} \sim p_F(\boldsymbol{\theta}^{(i)}|\boldsymbol{P}^{(i)};t)$ During training $t$ is sampled uniformly from $U(0,1)$.

**Training Strategy** We faced a similar representation—generation quality trade-off as described in ESM3 (Hayes et al., 2024) and DPLM (Wang et al., 2024). Intuitively a smaller $t$ would result in learning with lower quality input, whereas a larger $t$ would make the objective trivial. During training, for 90% of the time, we sample $t$ independently for each amino acid position, and for the remaining 10% of the time, the entire profile is trained with the same $t$. Additionally, as in DPLM (Wang et al., 2024), our backbone is first trained with masked language modeling objective.

### 3.3 FAMILY PROTEIN GENERATION

Given a protein family profile $\{\boldsymbol{P}^{(i)}\}_{i=1}^m \subset \Delta^{K-1}$, we first compute its Bayesian flow up to some initial time step $t_0$, then for $j$ in $[0, \cdots, N]$, $t_j \leftarrow \frac{(1-t_0)j}{N} + t_0$ do the following calculation iteratively:

$$\boldsymbol{\theta}_{t_j}^{(i)} \sim p_F(\boldsymbol{\theta}|\boldsymbol{P}_{\boldsymbol{\phi};j}^{(i)};t_j), \tag{10}$$

$$\boldsymbol{P}_{\boldsymbol{\phi};(j+1)} = f_{\boldsymbol{\phi}}(\boldsymbol{\theta}_{t_j}^{(1)}, \cdots, \boldsymbol{\theta}_{t_j}^{(m)}, t_j), \tag{11}$$

Where the initial $\{\boldsymbol{P}_{\boldsymbol{\phi};0}^{(i)}\}_{i=1}^m$ is set to $\{\boldsymbol{P}^{(i)}\}_{i=1}^m$. Finally we take the $\arg\max$ sampling over $\{\boldsymbol{P}_{\boldsymbol{\phi};(N+1)}^{(i)}\}_{i=1}^m$ to get the generated family protein sequence, the $i$th amino acid can be decoded as follows: $\mathrm{a}^{(i)} = \arg\max_k(\boldsymbol{P}_{\boldsymbol{\phi};(N+1)}^{(i)})_k$.

The initial time $t_0$ plays a critical role in the sampling process, setting a $t_0$ too small would lead to a severe loss of information from the conditioned sequence or family, while setting $t_0$ too large may limit the exploration of possible proteins. For individual protein sequences, we set $t_0$ to 0.3. However, profiles typically exhibit greater variance, necessitating a larger initial time step. In our experiments, we set the initial time step $t_0$ to 0.6 when sampling from a family profile.

## 4 EXPERIMENTS

In this section, we validate the advantages of ProfileBFN in family protein generation and protein representation learning through extensive experiments. In the following paragraphs, we present the outstanding performance results of ProfileBFN in family protein generation and protein representation learning tasks, and provide an in-depth analysis of these results. Finally, we analyze the sampling process of ProfileBFN, revealing its efficiency and the biological meaning inherent in this process.

A comprehensive overview of the training and evaluation configurations, including the metrics used, is provided in Appendix D.

### 4.1 MAIN RESULTS

**ProfileBFN Leads in Family Protein Generation** We collected 61 primary sequences released by CAMEO starting from May 4, 2024, and searched for their homologous sequences using the same procedure as described in AlphaFold2 (Jumper et al., 2021). The models, whether provided with a primary sequence or a set of homologous sequences, generate 1,000 sequences each for comparison. Refer to Appendix D.2.1 for more detailed information on the experimental settings and evaluation metrics.

Table 1 presents a comparison of the performance of different models in generating family proteins. Based on the results presented in the table, we provide the following analysis:

- From a structural perspective, Sequences belonging to the same family should share co-evolutionary information similar to that of the reference family. To evaluate this, we conducted non-parameterized contact prediction on the generated protein sets using the Potts

Table 1: Comparison of sequence and structural metrics (non-parametric cluster-level) on datasets collected from CAMEO. The results indicate that ProfileBFN outperforms in family protein generation.

| Model | Sequence | | Structure | | |
|---|---|---|---|---|---|
| | Div. ↓ | Nov. ↑ | LR P@L ↑ | LR P@L/2 ↑ | LR P@L/5 ↑ |
| Searched MSA | - | - | 0.186 | 0.270 | 0.395 |
| ESM-2 (150M) | 0.565 | **0.691** | 0.086 | 0.116 | 0.167 |
| ESM-2 (650M) | 0.619 | 0.556 | 0.100 | 0.146 | 0.223 |
| PoET-Single (201M) | 0.853 | 0.200 | 0.025 | 0.028 | 0.031 |
| PoET-MSA (201M) | 0.651 | 0.243 | 0.036 | 0.042 | 0.051 |
| EvoDiff-MSA (100M) | **0.225** | 0.668 | 0.061 | 0.089 | 0.168 |
| DPLM (150M) | 0.369 | 0.463 | 0.093 | 0.147 | 0.284 |
| DPLM (650M) | 0.445 | 0.411 | 0.102 | 0.159 | 0.303 |
| ProfileBFN-Single (150M) | 0.368 | 0.646 | 0.126 | 0.197 | 0.321 |
| ProfileBFN-Single (650M) | 0.421 | 0.581 | 0.162 | 0.262 | 0.422 |
| ProfileBFN-Profile (150M) | 0.283 | 0.650 | 0.128 | 0.210 | 0.384 |
| ProfileBFN-Profile (650M) | 0.293 | 0.641 | **0.173** | **0.280** | **0.474** |

model implemented in CCMPred. the LR P@L, LR P@L/2, LR P@L/5 is the precision at L, L/2, and L/5, respectively. This approach was chosen because parameterized models such as ESMFold or AlphaFold are prone to hallucination issues, as demonstrated in Appendix D.2.2.

ProfileBFN shows a considerable advantage, with its performance metrics even surpassing those of MSA obtained through search methods (see the first and last rows). This finding suggests that the sequences generated by ProfileBFN effectively capture the structural characteristics of the family, an example illustrating this is provided in Figure 1.

- From the perspective of sequence analysis, we expect the generated sequences to exhibit adequate diversity and novelty. To measure diversity, we use the mean identity value among the generated sequences, denoted as **Div**. Novelty is assessed by calculating the maximum identity between the generated sequences and natural sequences, with novelty defined as $1 - \max(\text{identity})$ and denoted as **Nov**.

  ProfileBFN excels in terms of diversity and novelty. These results indicate that ProfileBFN can generate diverse and novel sequences without suffering from severe mode collapse and ensures the production of varied outputs.

- Compared to our diffusion competitor, DPLM, ProfileBFN consistently outperforms across all metrics with significantly better results at different model sizes (rows 7, 8 vs 9, 10). This demonstrates the superiority of BFN in handling discrete variables over diffusion models.

- Comparing the performance of ProfileBFN models in different sizes, larger models generally capture family structure characteristics more effectively. However, they show a slight decline in performance regarding diversity and novelty. This is primarily due to the antagonistic relationship between structural conservativeness and sequence diversity and novelty.

- Regarding the input types for ProfileBFN, utilizing a profile derived from multiple sequence alignment (MSA) as input offers superior structural performance compared to a single sequence, while also enhancing diversity and novelty. This is because the profile or MSA contains richer structural information and more accurately reflects conservation across different sites. As a result, the model can more effectively capture structural features while ensuring diversity and novelty by modifying the more flexible sites.

Following PoET (Truong Jr & Bepler, 2023), we also use the structure prediction model ESMFold to evaluate the performance of different models (see Figure 5). Based on the results shown in the figure, the sequences generated by ProfileBFN exhibit higher pLDDT and Max TM-score values, indicating that ProfileBFN still holds an advantage in capturing structural conservation at the instance-level metrics. In terms of novelty, ProfileBFN ranks in the middle, but still offers a sufficient number of novel options. In contrast, while EvoDiff excels in diversity, it does not effectively capture the structural conserved features of the family. Overall, ProfileBFN still delivers the best performance in

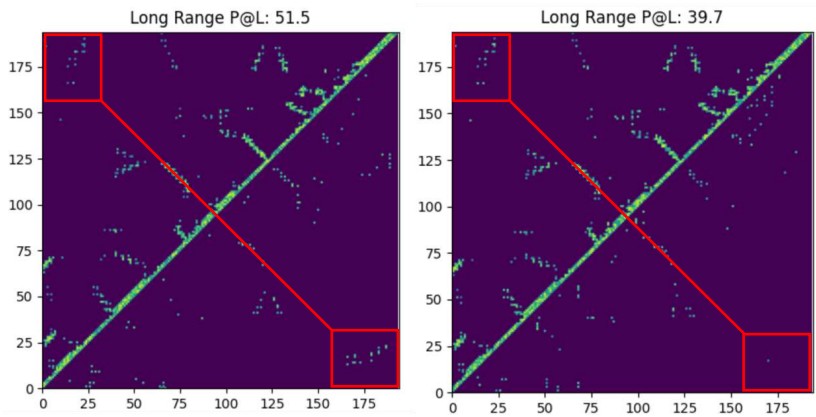

Figure 1: Example of contact map obtained using ProfileBFN (left) and Searched MSA (right). The family sequences generated by ProfileBFN even achieve more accurate predictions than the Searched MSA.

family protein generation. We provide three cases generated by ProfileBFN in Figure 2. However, it is important to note that the parameterized instance-level metrics have significant flaws, we provide further discussion in the Appendix D.2.2.

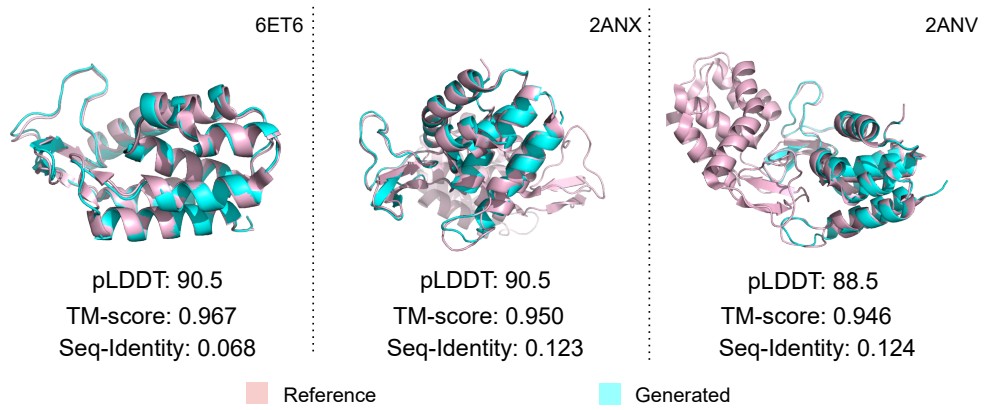

Figure 2: Three structurally conserved but sequence-novel lysozymes are generated by ProfileBFN.

**ProfileBFN Generates Functional Proteins** We utilize the enzyme function prediction model, CLEAN (Yu et al., 2023), to classify and evaluate enzymes generated by multiple models. Specifically, we focus on three representative categories of catalytic enzymes, each extensively validated experimentally. Models in consideration generate new enzymes based on reference sequences from each category. Subsequently, we use CLEAN to predict the EC numbers of these generated enzymes, thereby assessing their catalytic activity. Refer to Appendix E.1 for detailed information on the experimental settings and evaluation metrics.

From the results in the table 2, we measure Accuracy × Uniqueness, more extensive results are shown in Table 6, we can observe that the enzymes generated by ProfileBFN are considered more likely to possess the corresponding functions. From a functional perspective, ProfileBFN provides the best capability for generating family proteins.

From Table 6, PoET achieves the highest accuracy among all models. However, it suffers from mode collapse, leading to relatively low performance when evaluated using the combined metric of Accuracy × Uniqueness. In contrast, both ProfileBFN and EvoDiff generate a variety of results without observing mode collapse.

Table 2: Performance on enzyme tasks. We report the Accuracy $\times$ Uniqueness metric, complementary results can be found in Table 6. The results show that the enzymes generated by ProfileBFN are likely to be considered as having corresponding functions.

| Model | P40925 ↑ | Q7X7H9 ↑ | Q15165 ↑ |
|---|---|---|---|
| PoET-MSA | 3.00% | 33.3% | 0.05% |
| EvoDiff-MSA | 27.93% | 88.69% | 1.39% |
| ProfileBFN-Profile (650M) | **95.19%** | **98.98%** | **42.67%** |

**ProfileBFN Understands Proteins Deeply** To evaluate ProfileBFN's ability to represent proteins, we assess its performance on several protein prediction tasks (Wang et al., 2024; Su et al., 2023; Dallago et al., 2021), including protein function prediction (thermostability and metal ion binding), localization prediction (DeepLoc), annotation prediction (EC and GO), and protein-protein interaction prediction (HumanPPI). Following DPLM (Wang et al., 2024), we conduct full-parameter supervised fine-tuning on each dataset.

We use accuracy (**ACC%**) as the primary evaluation metric for most representation learning tasks. For thermostability, we compute Spearman's correlation (**Spearman's** $\rho$) (Zar, 2005), and for EC and GO annotation tasks, we use the maximum F1-score (**Fmax**). Refer to Appendix D.2.3 for detailed metric description.

Table 3 shows the performance of different models across various prediction tasks. Based on the results in the table, ProfileBFN outperforms its discrete diffusion competitor, DPLM, across all task metrics (see the last four rows).[3] The improvement in performance is attributed to the smoother data denoising process of BFN compared to discrete diffusion, as well as the removal of the adverse impact of unnatural MASK tokens on protein data. Specifically, BFN takes into account changes in the probability distribution of amino acid types at different positions, enabling the model to learn more detailed information about amino acid co-variation (e.g., the probability of amino acid types at two positions increasing or decreasing simultaneously). In contrast, discrete diffusion only considers changes in amino acid types (i.e., both positions undergo a type switch), resulting in a coarser granularity of model learning. Moreover, the BFN framework eliminates the need to introduce artificial MASK tokens, avoiding inconsistencies between upstream training and downstream tasks. The benefits of removing the MASK token have also been reported in the field of natural language processing (Yang, 2019).

Moreover, ProfileBFN demonstrates comparable performance to SaPort (Su et al., 2023), which explicitly utilizes protein structure information. This indicates that ProfileBFN has also developed a profound understanding of protein structure through learning from a large volume of protein sequences. However, it should also be noted that for tasks directly related to structural information, such as HumanPPI, SaProt still maintains a leading position, suggesting the necessity of integrating structural information into ProfileBFN in future work.

## 4.2 SAMPLING PROCESS ANALYSIS

In this section, we analyze the sampling process of ProfileBFN, including sampling efficiency and the biological meaning implied in the sampling process.

**ProfileBFN Achieves Higher Sampling Efficiency** Figure 3 shows a comparison of sampling times for different models when generating a protein of varying lengths. As observed from the figure, across different model sizes and protein lengths, ProfileBFN consistently demonstrates higher sampling efficiency compared to our main competitor, DPLM. Moreover, this advantage becomes more pronounced as the protein length increases. Although both DPLM and ProfileBFN utilize a similar ESM-2 network backbone, the need for a resampling trick (where each sampling step requires the model to infer twice) during DPLM's sampling process leads to a significant difference in their sampling efficiency. Compared to ESM-2, ProfileBFN incurs a slight loss in efficiency.

---

[3]According to the results provided by Wang et al. (2024), the performance of ProfileBFN and DPLM is comparable, with each having its own strengths and weaknesses. However, based on our replication experiments, ProfileBFN consistently outperforms DPLM. This may be attributed to the unstable training process of DPLM.

Table 3: Performance on various protein prediction tasks. ProfileBFN shows a strong understanding of proteins. *: protein structure is provided. †: results are quoted from SaProt (Su et al., 2023). ♡: results are quoted from DPLM (Wang et al., 2024). ⋄: results are reproduced by us using the official code and data. Our model is compared with the ⋄ version of the baseline models, if multiple versions exist.

| Model | Thermostability | HumanPPI | Metal Ion Binding | EC | GO | | | DeepLoc | |
|---|---|---|---|---|---|---|---|---|---|
| | | | | | MF | BP | CC | Subcellular | Binary |
| | Spearman's $\rho$ | ACC(%) | ACC(%) | Fmax | Fmax | Fmax | Fmax | ACC(%) | ACC(%) |
| SaProt* † | 0.724 | 86.41 | 75.75 | 0.884 | 0.678 | 0.356 | 0.414 | 85.57 | 93.55 |
| MIF-ST* † | 0.694 | 75.54 | 75.08 | 0.803 | 0.627 | 0.239 | 0.248 | 78.96 | 91.76 |
| ESM-1 (1B) † | 0.708 | 82.22 | 73.57 | 0.859 | 0.661 | 0.320 | 0.392 | 80.33 | 92.83 |
| ESM-2 (650M) † | 0.680 | 76.67 | 71.56 | 0.877 | 0.668 | 0.345 | 0.411 | 82.09 | 91.96 |
| AR-LM (650M) ♡ | 0.638 | 68.48 | 61.16 | 0.691 | 0.566 | 0.258 | 0.287 | 68.53 | 88.31 |
| DPLM (650M) ♡ | 0.695 | 86.41 | 75.15 | 0.875 | 0.680 | 0.357 | 0.409 | 84.56 | 93.09 |
| DPLM (650M) ⋄ | 0.698 | 77.77 | 70.52 | 0.881 | 0.659 | 0.330 | 0.388 | 85.98 | 93.17 |
| ProfileBFN (650M) | **0.710** | **82.22** | **74.58** | **0.887** | **0.673** | **0.342** | **0.416** | **86.80** | **93.58** |
| DPLM (150M) † | 0.687 | 80.98 | 72.17 | 0.822 | 0.662 | 0.328 | 0.379 | 82.41 | 92.63 |
| ProfileBFN (150M) | **0.701** | 78.88 | **77.74** | **0.874** | **0.672** | **0.341** | **0.394** | **82.73** | **93.52** |

However, this gap narrows as the model size and protein length decrease. Notably, EvoDiff, which has the fewest model parameters, exhibits the lowest sampling efficiency. This is because the model requires MSA as an input for family design. When designing proteins of the same length, the actual input size for the model is larger, leading to higher computational complexity.

**Sampling Process Reflects Protein Conservation** The sampling process of ProfileBFN is essentially a transition from a high entropy state to a low entropy state. In this paragraph, we explore the relationship between this process and the conservation of different protein sites. Specifically, we sum the entropy at each time step during the sampling process of ProfileBFN and compare it with the results of the site conservation analysis using MSA. Figure 4 presents an example of the extent of variability in the ProfileBFN sampling process alongside conserved protein sites analyzed through MSA (lysozyme Q37875). The figure shows a high consistency between the variation intensity at different sites during the sampling process and the conserved protein sites identified by MSA analysis. This indicates that ProfileBFN successfully captures the variability and conservation of different sites on the protein, and during the sampling process, it reflects this by controlling the extent of amino acid variation at these sites.

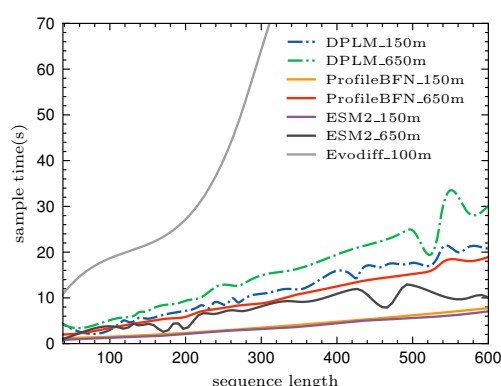

Figure 3: Sampling efficiency comparison. ProfileBFN has a higher sampling efficiency compared to its competitors.

# 5 RELATED WORK

**De novo protein design methods** constructs entirely new protein sequences that do not based on homologs. It perform self-supervised learning from large protein databases (Consortium, 2015; Mirdita et al., 2017; Suzek et al., 2007), aiming to model the evolutionary constraints across various families (Koonin et al., 2004; Meier et al., 2021; Lin et al., 2022). It demonstrates its advantages in scenarios of designing proteins for entirely new properties, especially in cases where there is limited homologous information (Madani et al., 2020; Nijkamp et al., 2023; Meng et al., 2023). However, it performs poorly in tasks involving the design of new proteins within large protein families.

**Mutation-based directed evolution approach** mimics the process of protein evolution. By training on evolutionary-scale protein sequences, it can capture key sites in protein evolution and model

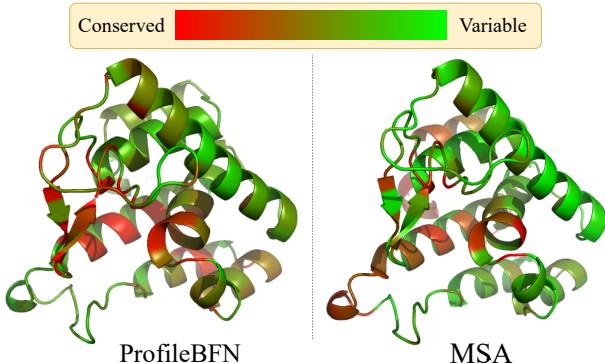

Figure 4: ProfileBFN's sampling process implies the conservation of proteins.

protein evolution process (Alamdari et al., 2023; Wang et al., 2024; Watson et al., 2023). It can design proteins that can be verified by wet experiments, but limited by the way of evolution from wild-type, so they cannot generate diverse protein sequences for reaching the optimal proteins.

**Protein family design**  is a protein design process that models homologous protein sequences as additional signals. These models can be further categorized into autoregressive and non-autoregressive models. For example, PoET (Truong Jr & Bepler, 2023), MSAGPT (Chen et al., 2024), and ProtMamba (Sgarbossa et al., 2024) are autoregressive models that take sequentially concatenated sequences as input and generate new proteins autoregressively. In contrast, EvoDiff-MSA (Alamdari et al., 2023) uses MSA-Transformer (Rao et al., 2021) as its MSA module, takes an MSA matrix as input, and generates new proteins in a non-autoregressive manner.

## 6 CONCLUSION

In this paper, we have made significant contributions to the field of protein sequence generation with several key advancements. We extended the Discrete BFN to design the ProfileBFN model, which effectively utilizes protein family profile information for generating family-specific protein sequences. Through formal derivation, we introduced a new Bayesian flow and loss component, making the ProfileBFN versatile and applicable to any data with profile characteristics.

Our ProfileBFN model can accommodate both single-sequence and multiple-sequence profiles. This flexibility allows us to train on single-sequence data while generating sequences using multi-sequence profiles, thus avoiding the costly process of constructing profile training datasets.

Our model demonstrated exceptional performance in both representation and generation tasks. The generated sequences showed biologically meaningful variations in the amino acid positions, which is crucial for practical applications in protein engineering and functional analysis.

Overall, our proposed ProfileBFN have exhibited robustness, efficiency, and biological relevance, offering a promising tool for protein sequence generation and functional studies.

## ETHICS STATEMENT

In conducting our research on the Profile Bayesian Flow Networks (ProfileBFN) for generative modeling of protein families, we have adhered to the highest ethical standards and address potential concerns as follows:

1. **Data Use and Privacy** Our research did not involve human subjects or private data. All protein sequence data used in our experiments were obtained from publicly available databases, which are free for academic and scientific research use. No identifiable personal data were used or generated.

2. **Potentially Harmful Insights and Applications** The development of protein design technologies, including our proposed ProfileBFN, has the potential for beneficial applications in fields such as medicine, bioengineering, and environmental science.

3. **Bias and Fairness** We have taken steps to ensure that our model and methodologies do not inadvertently introduce bias in the generated protein sequences. The ProfileBFNmodel is designed to be applicable to a wide variety of protein families without favoring any particular family or type. We emphasize the importance of continued evaluation and validation to maintain fairness and accuracy in diverse biological applications.

4. **Environmental Impact** To minimize our environmental footprint, we optimized computational resources by training on single-sequence data, thereby avoiding the need for large-scale MSA data construction and reducing computational power consumption. This approach also contributes to the sustainability of scientific research practices.

5. **Research Integrity** We uphold the principles of scientific integrity and transparency in our research. All methods and results have been meticulously documented. We encourage reproducibility by providing detailed descriptions of our algorithms and experiments, facilitating validation by other researchers.

In conclusion, while the potential applications of ProfileBFN offer significant advancements in protein design, we remain committed to conducting our research ethically and responsibly, with careful consideration of potential implications and societal impacts.

## ACKNOWLEDGEMENTS

This work is supported by the Natural Science Foundation of China (Grant No. 62376133) , Wuxi Research Institute of Applied Technologies, Tsinghua University under Grant 20242001120 and sponsored by Beijing Nova Program (20240484682).

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

## A    PROFILE BFN DERIVATION

### A.1    THE ESSENCE OF BAYESIAN FLOW NETWORKS

For easy understanding, the reader can treat the variables as discrete variables. Without loss of generality, the formulation can be easily extended to continuous variables by swapping the summation with integration. This section reviews the essence of Bayesian Flow Networks (BFN)(Graves et al., 2023) in a more simple language, there is a defined noisy channel $q(\cdot|\mathrm{x};\omega)$, through which a variable x leaks it's information $z_i \sim q(\cdot|\mathrm{x};\omega)$. An observer then receives the leaked information and updates its belief about the variable x through Bayesian update and obtain a belief about x: $p(\mathrm{x}|z_{1:n})$.

In a bits-back coding scheme, the total nats required to transfer x with $z_{1:n}$ as intermediate latent is $-\log p(z_{1:n}) - \log p(x|z_{1:n})$ with $-\log q(z_{1:n}|x)$ nats put back, so the expected marginal nats required to transfer data from $p(x)$ is:

$$\mathbb{E}_{p(x)}\mathbb{E}_{q(z_{1:n}|x;\omega)}\left[-\log p(z_{1:n}) - \log p(x|z_{1:n}) + \log q(z_{1:n}|x;\omega)\right]$$

$$= \mathbb{E}_{p(x)}\left[\mathbb{E}_{q(z_{1:n}|x;\omega)}\log\frac{q(z_{1:n}|x;\omega)}{p(z_{1:n})} - \mathbb{E}_{q(z_{1:n}|x;\omega)}\log p(x|z_{1:n})\right] = -\text{VLB}$$

$$= \mathbb{E}_{p(x)}\left[D_{\text{KL}}(q(z_{1:n}|x;\omega)||p(z_{1:n})) - \mathbb{E}_{q(z_{1:n}|x;\omega)}\log p(x|z_{1:n})\right] \tag{12}$$

With the conditional distribution of the noisy channel $q(z_{1:n}|x)$ and a series of observed variables $z_{1:n}$, following bayesian update rule, the udpated belief of the variable x is:

$$q(x|z_{1:n}) = \frac{q(z_{1:n}|x)q(x)}{\sum_x q(z_{1:n}|x)q(x)} \tag{13}$$

There could be sparsity problem or curse of dimensionality problem when the variable x is high-dimensional. Thus $m$-dimensional x is treated as $m$ independent variables, and updated independently with bayesian update rule. To model the interdependence between variables, an neural network is introduced to rectify the posterior distribution $q(\cdot|z_{1:n}; \boldsymbol{\theta}^{(1)}, \cdots, \boldsymbol{\theta}^{(m)})$, where $\boldsymbol{\theta}^{(i)}$ is the governing parameter of the posterior distribution of the $i$-th component and determined by $z_{1:n}$.

$$p_{\boldsymbol{\phi}}(\cdot|z_{1:n}) = f_{\boldsymbol{\phi}}(q(\cdot|z_{1:n}; \boldsymbol{\theta}^{(1)}, \cdots, \boldsymbol{\theta}^{(m)})) \tag{14}$$

$$= f_{\boldsymbol{\phi}}(\boldsymbol{\theta}^{(1)}, \cdots, \boldsymbol{\theta}^{(m)}) \tag{15}$$

Without knowing x, variables $\{z_1, z_1, \cdots, z_n\}$ are correlated variables, $p(z_{1:n})$ in Eq. 12 is then factorized autoregressively as $p(z_{1:n}) = p(z_1)\prod_{i=2}^{n} p(z_i|z_{1:i-1})$, and further parameterized combining the output distribution $p_{\boldsymbol{\phi}}$ from the neural network in Eq. 14:

$$p(z_{1:n}) = p(z_1)\prod_{i=2}^{n} p(z_i|z_{1:i-1})$$

$$= \left(\sum_x q(z_1|x)p_{\boldsymbol{\phi}}(x|z_\emptyset)\right)\prod_{i=2}^{n}\sum_x q(z_i|x)p_{\boldsymbol{\phi}}(x|z_{1:i-1})$$

$$\overset{\text{def}}{=} \prod_{i=1}^{n} p_R(z_i|z_{1:i-1}; \boldsymbol{\phi}) \tag{16}$$

Plug Eq. 16 and Eq. 14 into Eq. 12, the $-\text{VLB}(\boldsymbol{\phi})$ is then:

$$-\text{VLB}(\boldsymbol{\phi}) = \mathbb{E}_{p(x)}\left[\sum_{i=1}^{n} D_{\text{KL}}(q(z_i|x;\omega)||p_R(z_i|z_{1:i-1}; \boldsymbol{\phi})) - \mathbb{E}_{q(z_{1:n}|x;\omega)}\log p_{\boldsymbol{\phi}}(x|z_{1:n})\right] \tag{17}$$

The $-\text{VLB}(\boldsymbol{\phi})$ is the expected marginal nats required to transfer a data from $p(x)$, with the transmission system parameterized by $\boldsymbol{\phi}$. The objective is to minimize the transmission cost, and the model is trained by minimizing the $-\text{VLB}(\boldsymbol{\phi})$.

## A.2 PROFILE BAYESIAN FLOW NETWORKS

In the original BFN paper, the continuous variable y is regarded as the latent variable that is used for data transmission, it treats each components of the categorical variable as binary variable from which y is derived with central limit theorem.

As it's not so straightforward to treat the continuous variable as the latent variable, and kind of "wrong" to treat the categorical variable as a set binary variables for derivation, we will derive the discrete Bayesian flow from a different perspective, where the latent variables $z_i$ are still the discrete evidences that are leaked from the data.

We arrive at the Theorem 3.1 that describes the continuous time discrete Bayesian flow with proper derivation and proof.

*Derivation and Proof of Theorem 3.1.* The noisy channel based on a profile is actually a generalization of the original BFN's noisy channel, where the profile is a one-hot vector. Exchanging the one-hot vector with a profile can be seen as a hierarchical sampling process: first sample a one-hot vector according to the profile, then do the same as the original BFN. Still, we consider the transmission of noisy samples $\{z_i\}_{i=1}^n$ as a sequential update of the belief of the variable x in the profile, and finally push $n \to +\infty$. Since sequential Bayesian update is equivalent to batch Bayesian update, $\forall$x:

$$p(\mathbf{x}|\mathbf{z}_{1:n}) = \frac{q(\mathbf{z}_n|\mathbf{x})p(\mathbf{x}|\mathbf{z}_{1:n-1})}{\sum_x q(\mathbf{z}_n|x)p(x|\mathbf{z}_{1:n-1})}$$

Define $\pi_i(\mathbf{x}) = p(\mathbf{x}|\mathbf{z}_{1:i})$, we have the following recursive form

$$\pi_i(\mathbf{x}) = \frac{q(\mathbf{z}_i|\mathbf{x})\pi_{i-1}(\mathbf{x})}{\sum_x q(\mathbf{z}_i|x)\pi_{i-1}(x)}$$

Where $q(\mathbf{z}_i|\mathbf{x}; \omega_i) = \frac{1-\omega_i}{K} + \omega_i \mathbf{1}_{\mathbf{z}_i=\mathbf{x}}$ is the one-hot noisy channel (the second hierarchy), $\omega_i$ omitted for brevity. After observing a new evidence $z_i$ the posterior distribution is:

$$\begin{aligned}
\pi_i(\mathbf{x}) &= \frac{\left(\frac{1-\omega_i}{K} + \omega_i \delta_{\mathbf{z}_i\mathbf{x}}\right)\pi_{i-1}(\mathbf{x})}{\sum_x \left(\frac{1-\omega_i}{K} + \omega_i \delta_{\mathbf{z}_i\mathbf{x}}\right)\pi_{i-1}(\mathbf{x})} \\
&= \frac{\left(\frac{1-\omega_i}{K} + \omega_i \delta_{\mathbf{z}_i\mathbf{x}}\right)\pi_{i-1}(\mathbf{x})}{\frac{1-\omega_i}{K} + \omega_i \pi_{i-1}(\mathbf{z}_i)}
\end{aligned}$$

where $\delta_{..}$ is the Kronecker delta function.

We then analyze how the observed evidence will affect the distribution in the log space, the accumulated log probability of the distribution is:

$$\begin{aligned}
\ln(\pi_i(\mathbf{x})) - \ln(\pi_{i-1}(\mathbf{x})) &= \ln\left(\frac{1-\omega_i}{K} + \omega_i \sigma_{\mathbf{z}_i\mathbf{x}}\right) + C \\
&= \begin{cases} \ln\left(\frac{1-\omega_i}{K} + \omega_i\right) + C & \mathbf{z}_i = \mathbf{x}, \\ \ln\left(\frac{1-\omega_i}{K}\right) + C & \mathbf{z}_i \neq \mathbf{x}, \end{cases}
\end{aligned}$$

where $C = -\ln\left(\frac{1-\omega_i}{K} + \omega_i \pi_{i-1}(\mathbf{z}_i)\right)$ is a constant that is irrelevant to x.

Notice that when observing an evidence $\mathbf{z}_i$ there will be an extra "energy" on the index matching the evidence by

$$\ln\left(\frac{1-\omega_i}{K} + \omega_i\right) - \ln\left(\frac{1-\omega_i}{K}\right) = \ln\left(1 + \frac{K\omega_i}{1-\omega_i}\right)$$

Following Graves et al. (2023) this term is defined as:

$$\ln \xi_i \stackrel{\text{def}}{=} \ln\left(1 + \frac{K\omega_i}{1-\omega_i}\right)$$

Below we assume all $\omega_i$'s are equal and simply denote $(\omega_i, \xi_i)$ as $(\omega, \xi)$.

Now we analyze the situation of having observed $m(m \leq n)$ evidences. Assume there are $c_x$ evidences observed for x such that $\sum_x c_x = m$, then the built up log probability for x after observing $m$ evidences is

$$\ln(\pi_m(\mathbf{x})) = c_x \ln \xi + \ln(\pi_0(\mathbf{x})) + C \tag{18}$$

The $c_x$'s are the counts of the evidences observed, which follow a multinomial distribution $\mathcal{M}(m, \frac{1-\omega}{K} + \omega\boldsymbol{\rho})$, so the expectation, variance and covariance of the counts are:

$$\mathbb{E}[c_x] = m\left(\frac{1-\omega}{K} + \omega\boldsymbol{\rho}_x\right)$$

$$\text{Var}[c_x] = m\left(\frac{1-\omega}{K} + \omega\boldsymbol{\rho}_x\right)\left(1 - \left(\frac{1-\omega}{K} + \omega\boldsymbol{\rho}_x\right)\right)$$

$$\text{Cov}[c_x, c_{x'}] = -m\left(\frac{1-\omega}{K} + \omega\boldsymbol{\rho}_x\right)\left(\frac{1-\omega}{K} + \omega\boldsymbol{\rho}_{x'}\right)(x \neq x')$$

Define $y_x = (c_x - m\frac{1-\omega}{K})\ln\xi$, the corresponding terms are:

$$\mathbb{E}[y_x] = m\omega\boldsymbol{\rho}_x \ln\xi$$

$$\text{Var}[y_x] = m\left(\frac{1-\omega}{K} + \omega\boldsymbol{\rho}_x\right)\left(1 - \left(\frac{1-\omega}{K} + \omega\boldsymbol{\rho}_x\right)\right)\ln^2\xi$$

$$\text{Cov}[y_x, y_{x'}] = -m\left(\frac{1-\omega}{K} + \omega\boldsymbol{\rho}_x\right)\left(\frac{1-\omega}{K} + \omega\boldsymbol{\rho}_{x'}\right)\ln^2\xi(x \neq x')$$

Note that $n \to +\infty \Rightarrow \omega \to 0$, the first order Taylor expansion of $\ln\xi$ is:

$$\ln\xi = \frac{K\omega}{1-\omega} + O(\omega^2)$$

According to the definition and assumption that all $\omega_i$'s are equal, $m\omega^2 = \beta(\frac{m}{n})$, so the expectation and covariance matrix of $\mathbf{y}$ are $\mathbb{E}[\mathbf{y}] = K\beta(\frac{m}{n})\boldsymbol{\rho}$ and $\beta(\frac{m}{n})\Sigma$, with $\Sigma_{ij} = K\mathbf{1}_{i=j} - 1$. As $n \to +\infty$, $\frac{m}{n}$ is replaced by $\beta(t)$ with a continuous time $t$.

We need to control the expected energy built up for each category to be bounded, thus $\beta(1)$ need to be bounded. □

As the latent variable for data transmission is different from the original BFN paper, the KL term in 17 should be rederived to fit the new setting. We propose Theorem 3.2 that describes the KL term in the continuous time discrete Bayesian flow with proper derivation and proof.

*Derivation and Proof of Theorem 3.2.*

$$\lim_{n\to+\infty} nD_{\text{KL}}(q(\mathbf{z}|\boldsymbol{\rho})||p(\mathbf{z}))$$

$$= \lim_{n\to+\infty}\left(n\sum_{\mathbf{z}} q(\mathbf{z}|\boldsymbol{\rho})\log q(\mathbf{z}|\boldsymbol{\rho}) - n\sum_{\mathbf{z}} q(\mathbf{z}|\boldsymbol{\rho})\log p(\mathbf{z})\right) \tag{19}$$

Rearrange the inner right term in Eq. 19:

$$n\sum_{\mathbf{z}} q(\mathbf{z}|\boldsymbol{\rho})\log p(\mathbf{z})$$

$$= n\sum_{\mathbf{z}} q(\mathbf{z}|\boldsymbol{\rho})\log\left(\frac{1-\omega}{K} + p_\phi(\mathbf{z})\omega\right)$$

$$= \sum_{\mathbf{z}} -nq(\mathbf{z}|\boldsymbol{\rho})\log K + \sum_{\mathbf{z}} nq(\mathbf{z}|\boldsymbol{\rho})\log(1 + (Kp_\phi(\mathbf{z}) - 1)\omega) \tag{20}$$

Apply second order Taylor expansion on the right term in Eq. 20:

$$\sum_{\mathbf{z}} nq(\mathbf{z}|\boldsymbol{\rho})\log(1 + (Kp_\phi(\mathbf{z}) - 1)\omega)$$

$$= \sum_{\mathbf{z}} nq(\mathbf{z}|\boldsymbol{\rho})\left((Kp_\phi(\mathbf{z}) - 1)\omega - \frac{1}{2}(Kp_\phi(\mathbf{z}) - 1)^2\omega^2 + o(\omega^3)\right)$$

$$= \sum_{\mathbf{z}} nq(\mathbf{z}|\boldsymbol{\rho})(Kp_\phi(\mathbf{z}) - 1)\omega - \frac{1}{2}\sum_{\mathbf{z}} nq(\mathbf{z}|\boldsymbol{\rho})(Kp_\phi(\mathbf{z}) - 1)^2\omega^2 + o(1) \tag{21}$$

The first term in Eq. 21 can be expanded as:

$$\sum_{\mathbf{z}} n q(\mathbf{z}|\boldsymbol{\rho})(Kp_\phi(\mathbf{z}) - 1)\omega$$

$$= \sum_{\mathbf{z}} n \left(\frac{1-\omega}{K} + \omega\boldsymbol{\rho}(\mathbf{z})\right)(Kp_\phi(\mathbf{z}) - 1)\omega$$

$$= \sum_{\mathbf{z}} n \frac{1-\omega}{K}(Kp_\phi(\mathbf{z}) - 1)\omega + n\omega^2 \sum_{\mathbf{z}} \boldsymbol{\rho}(\mathbf{z})(Kp_\phi(\mathbf{z}) - 1)$$

$$= 0 + \beta \sum_{\mathbf{z}} \boldsymbol{\rho}(\mathbf{z})(Kp_\phi(\mathbf{z}) - 1) = \beta \sum_{\mathbf{z}} \boldsymbol{\rho}(\mathbf{z})(Kp_\phi(\mathbf{z}) - 1) \tag{22}$$

The second term in Eq. 21 can be expanded as:

$$\sum_{\mathbf{z}} n q(\mathbf{z}|\boldsymbol{\rho})(Kp_\phi(\mathbf{z}) - 1)^2\omega^2$$

$$= \sum_{\mathbf{z}} \beta \left(\frac{1-\omega}{K} + \omega\boldsymbol{\rho}(\mathbf{z})\right)(Kp_\phi(\mathbf{z}) - 1)^2$$

$$= \sum_{\mathbf{z}} \beta \frac{1-\omega}{K}(Kp_\phi(\mathbf{z}) - 1)^2 + \beta\omega \sum_{\mathbf{z}} \boldsymbol{\rho}(\mathbf{z})(Kp_\phi(\mathbf{z}) - 1)^2$$

$$= \sum_{\mathbf{z}} \beta \frac{1-\omega}{K}(K^2 p_\phi^2(\mathbf{z}) + 1 - 2Kp_\phi(\mathbf{z})) + \beta\omega \sum_{\mathbf{z}} \boldsymbol{\rho}(\mathbf{z})(Kp_\phi(\mathbf{z}) - 1)^2$$

$$= -\beta + \beta K ||p_\phi||^2 + o(1) \tag{23}$$

Plug Eq. 22 and Eq. 23 into Eq. 21, and Eq. 21 into Eq. 20, and Eq. 20 becomes:

$$n \sum_{\mathbf{z}} q(\mathbf{z}|\mathbf{x}) \log p_\phi(\mathbf{z})$$

$$= -n \log K + \beta \sum_{\mathbf{z}} \boldsymbol{\rho}(\mathbf{z})(Kp_\phi(\mathbf{z}) - 1) - \frac{1}{2}\left(-\beta + \beta K ||p_\phi||^2\right) + o(1)$$

$$= -n \log K + \beta K \sum_{\mathbf{z}} \boldsymbol{\rho}(\mathbf{z})p_\phi(\mathbf{z}) - \frac{1}{2}\beta - \frac{1}{2}\beta K ||p_\phi||^2 \tag{24}$$

Similarly, plug the $p_\phi$ with $\boldsymbol{\rho}$, into Eq. 24, then the first term in Eq. 19 can be transformed to:

$$n \sum_{\mathbf{z}} q(\mathbf{z}|\mathbf{x}) \log q(\mathbf{z}|\boldsymbol{\rho})$$

$$= -n \log K + \frac{1}{2}\beta K ||\boldsymbol{\rho}||^2 - \frac{1}{2}\beta + o(1) \tag{25}$$

Since $\beta = n\omega^2$ is bounded as $n \to +\infty$, the $o(1)$ term is negligible. Plug Eq. 24 and Eq. 25 into Eq. 19, we have:

$$\lim_{n \to +\infty} n D_{\mathrm{KL}}(q(\mathbf{z}|\boldsymbol{\rho}) || p(\mathbf{z})) = \frac{1}{2}\beta K ||p_\phi - \boldsymbol{\rho}||^2 \tag{26}$$

For $\omega(t)$ that satisfies $\beta(t) = \int_0^t \omega^2(\tau)d\tau, 1 \geq t \geq 0, \beta(1) = const$, the limit of the KL divergence can be easily derived by the same method with the following substitution:

$$\omega \to \sqrt{\beta'(t)\Delta t}, \tag{27}$$

$$n \to \frac{1}{\Delta t}, \tag{28}$$

$$n\omega^2 \to \beta'(t), \tag{29}$$

$$\lim_{n \to +\infty} \to \lim_{\Delta t \to 0} \tag{30}$$

and the resulted KL divergence is:

$$\lim_{n \to +\infty} n D_{\mathrm{KL}}(q(\mathbf{z}|\boldsymbol{\rho};t)||p(\mathbf{z};t)) \tag{31}$$

$$= \lim_{\Delta t \to 0} \frac{1}{\Delta t} D_{\mathrm{KL}}(q(\mathbf{z}|\boldsymbol{\rho};t)||p(\mathbf{z};t)) = \frac{1}{2}\beta'(t)K||p_{\boldsymbol{\phi}} - \boldsymbol{\rho}||^2 \tag{32}$$

$$\square$$

It seems although starting from a different perspective from the original BFN paper, the derived KL term arrived at the same form as the original BFN paper.

As the reconstruction term in the right of Eq. 17 will trivially approach 0 when $\beta(1)$ is sufficiently large, the training loss for some $\boldsymbol{\rho}$ at some step $t$ is then:

$$\mathcal{L}(\mathbf{x}) = \frac{1}{2}\beta'(t)K||p_{\boldsymbol{\phi}} - \boldsymbol{\rho}||^2 \tag{33}$$

## B  ALGORITHMS

---
**Algorithm 1** Training Loss Procedure

---
**Require:** $\beta_1 \in \mathbb{R}$, vocabulary size $K \in \mathbb{Z}^+$, a neural network $f_{\boldsymbol{\phi}}(\boldsymbol{\theta}^{(1)}, \cdots, \boldsymbol{\theta}^{(m)}, t)$, where $\boldsymbol{\phi}$ is the parameter of the neural network.
**Input:** profiles $\{\boldsymbol{P}^{(i)}\}_{i=1}^m \subset \Delta^{K-1}$, where $m$ is the sequence length
$t \sim U(0,1)$
$\beta_t \leftarrow t\beta_1$
$\mathbf{y}_t^{(i)} \sim \mathcal{N}(K\beta_t \boldsymbol{P}^{(i)}, \beta_t \mathcal{C})$
$\boldsymbol{\theta}_t^{(i)} = \mathrm{softmax}(\mathbf{y}_t^{(i)})$
$\{\boldsymbol{P}_{\boldsymbol{\phi}}^{(i)}\}_{i=1}^m = f_{\boldsymbol{\phi}}(\boldsymbol{\theta}_t^{(1)}, \cdots, \boldsymbol{\theta}_t^{(m)}, t)$
$\mathcal{L}(\boldsymbol{P}) = \sum_{i=1}^m \frac{1}{2}\beta_1 K||(\boldsymbol{P}_{\boldsymbol{\phi}}^{(i)} - \boldsymbol{P}^{(i)}||^2$
**Return** $\mathcal{L}(\boldsymbol{P})$

---

---
**Algorithm 2** Family Protein Generation Procedure

---
**Require:** $\beta_1 \in \mathbb{R}$, vocabulary size $K \in \mathbb{Z}^+$, initial time $t_0$, sampling steps $N$,
a neural network $f_{\boldsymbol{\phi}}(\boldsymbol{\theta}^{(1)}, \cdots, \boldsymbol{\theta}^{(m)}, t)$, where $\boldsymbol{\phi}$ is the parameter of the neural network.
**Input:** profiles $\{\boldsymbol{P}^{(i)}\}_{i=1}^m \subset \Delta^{K-1}$ of certain protein family, where $m$ is the sequence length.
**for** $j = 0$ to $N$ **do**
$\quad t \leftarrow \frac{(1-t_0)j}{N} + t_0$
$\quad \beta_t \leftarrow t\beta_1$
$\quad \mathbf{y}^{(i)} \sim \mathcal{N}(K\beta_t \boldsymbol{P}^{(i)}, \beta_t \mathcal{C})$
$\quad \boldsymbol{\theta}^{(i)} = \mathrm{softmax}(\mathbf{y}^{(i)})$
$\quad \{\boldsymbol{P}^{(i)}\}_{i=1}^m = f_{\boldsymbol{\phi}}(\boldsymbol{\theta}^{(1)}, \cdots, \boldsymbol{\theta}^{(m)}, t)$
**end for**
$\mathbf{a}^{(i)} \leftarrow \arg\max_k (\boldsymbol{P}^{(i)})_k$
**Return** $\{\mathbf{a}^{(i)}\}_{i=1}^m$

---

## C  DATASETS

### C.1  EVALUATION DATASETS

Three datasets were used to evaluate the performance of our model of protein family generation: dataset from CAMEO, enzyme families, and phage lysozyme families. The dataset collected from CAMEO, which contains 61 proteins with Homo-oligomer Assessment as detailed in Table 4, was

Table 4: Detailed information of each protein for CAMEO dataset.

| ID | PDB | Chain | Length | Title |
|---|---|---|---|---|
| 1 | 8BL5 | A | 148 | Crystal Structure of Sam0.26 |
| 2 | 8F9Q | A | 505 | Guinea pig sialic acid esterase |
| 3 | 8F9R | A | 501 | Rabbit sialic acid esterase |
| 4 | 8FSL | C | 116 | Human Mesothelin bound to a neutralizing VH domain antibody |
| 5 | 8HJP | A | 459 | Crystal structure of glycosyltransferase SgUGT94-289-3 in complex with UDP state 1 |
| 6 | 8ISO | A | 269 | Crystal structure of extended-spectrum class A beta-lactamase, CESS-1 |
| 7 | 8IXT | A | 427 | Rat Transcobalamin in Complex with Glutathionylcobalamin |
| 8 | 8JDH | A | 166 | Crystal structure of anti-CRISPR AcrIF25 |
| 9 | 8JGO | A | 535 | Crystal structure of Deinococcus radiodurans exopolyphosphatase |
| 10 | 8JI1 | A | 198 | Crystal structure of Ham1 from Plasmodium falciparum |
| 11 | 8JIJ | A | 421 | Alanine decarboxylase |
| 12 | 8JJA | A | 216 | SP1746 in complex with acetate ions |
| 13 | 8JRB | A | 597 | Structure of DNA polymerase 1 from Aquifex pyrophilus |
| 14 | 8JYX | A | 635 | Crystal structure of the gasdermin-like protein RCD-1-1 from Neurospora crassa |
| 15 | 8K05 | A | 340 | Pseudouridine 5-monophosphate glycosylase from Arabidopsis thaliana – sulfate bound holoenzyme |
| 16 | 8K40 | A | 456 | mercuric reductase,GbsMerA, - FAD bound |
| 17 | 8OV9 | A | 350 | Crystal structure of Ene-reductase 1 from black poplar mushroom |
| 18 | 8OXR | A | 145 | Structure of the N-terminal didomain d1-d2 of the Thrombospondin type-1 domain-containing 7A |
| 19 | 8OYD | A | 45 | TrkB transmembrane domain NMR structure in DMPC/DHPC bicelles |
| 20 | 8OZZ | A | 114 | PH domain of AKT-like kinase in Trypanosoma cruzi |
| 21 | 8PIH | C | 118 | Structure of Api m1 in complex with two nanobodies |
| 22 | 8QL0 | A | 693 | Structure of human PAD6 Phosphomimic mutant V10E/S446E, apo |
| 23 | 8QLC | A | 627 | Crystal structure of the pneumococcal Substrate-binding protein AliD in open conformation |
| 24 | 8QLH | A | 633 | Crystal structure of the pneumococcal Substrate-binding protein AliC as a domain-swapped dimer |
| 25 | 8QPM | A | 100 | Structure of methylene-tetrahydromethanopterin reductase from Methanocaldococcus jannaschii |
| 26 | 8QQ5 | A | 222 | Structure of WT SpNox DH domain: a bacterial NADPH oxidase. |
| 27 | 8QVC | B | 100 | Deinococcus aerius TR0125 C-glucosyl deglycosidase (CGD), wild type crystal cryoprotected with glycerol |
| 28 | 8QZ1 | C | 136 | Crystal structure of human two pore domain potassium ion channel TREK-2 (K2P10.1) in complex with a nanobody (Nb58) |
| 29 | 8QZ2 | C | 134 | Crystal structure of human two pore domain potassium ion channel TREK-2 (K2P10.1) in complex with an inhibitory nanobody (Nb61) |
| 30 | 8QZ3 | C | 137 | Crystal structure of human two pore domain potassium ion channel TREK-2 (K2P10.1) in complex with an activatory nanobody (Nb67) |
| 31 | 8R3R | A | 673 | Transketolase from Streptococcus pneumoniae in complex with thiamin pyrophosphate |
| 32 | 8R3S | A | 677 | Transketolase from Staphylococcus aureus in complex with thiamin pyrophosphate |
| 33 | 8R8O | A | 275 | Hallucinated de novo TIM barrel with three helical extensions - HalluTIM3-1 |
| 34 | 8S4S | A | 145 | PrgE from plasmid pCF10 |
| 35 | 8SUC | A | 100 | NHL-2 NHL domain |
| 36 | 8SUF | A | 1007 | The complex of TOL-1 ectodomain bound to LAT-1 Lectin domain |
| 37 | 8SUF | A | 114 | The complex of TOL-1 ectodomain bound to LAT-1 Lectin domain |
| 38 | 8SW5 | C | 47 | Protein Phosphatase 1 in complex with PP1-specific Phosphatase targeting peptide (PhosTAP) version 1 |
| 39 | 8TB2 | A | 100 | Structure of SasG (type II) (residues 165-421) from Staphylococcus aureus MW2 |
| 40 | 8TI6 | A | 155 | Crystal structure of Tyr p 36.0101 |
| 41 | 8UAI | B | 494 | Crystal structure of hetero hexameric hazelnut allergen Cor a 9 |
| 41 | 8UAI | D | 493 | Crystal structure of hetero hexameric hazelnut allergen Cor a 9 |
| 43 | 8V8L | A | 237 | Switchgrass Chalcone Isomerase |
| 44 | 8V8P | A | 231 | Sorghum Chalcone Isomerase |
| 45 | 8W1D | A | 177 | CRYSTAL STRUCTURE OF DPS-LIKE PROTEIN PA4880 FROM PSEUDOMONAS AERUGINOSA (DIMERIC FORM) |
| 46 | 8W6V | A | 536 | Structural basis of chorismate isomerization by Arabidopsis isochorismate synthase ICS1 |
| 47 | 8W26 | A | 429 | X-ray crystal structure of the GAF-PHY domains of SyB-Cph1 |
| 48 | 8W53 | B | 488 | Crystal structure of LbUGT in complex with UDP |
| 49 | 8WEX | A | 468 | Crystal structure of N-acetyl sugar amidotransferase from Legionella pneumophila |
| 50 | 8WG0 | D | 100 | Crystal structure of GH97 glucodextranase from Flavobacterium johnsoniae in complex with glucose |
| 51 | 8WOP | A | 100 | Crystal structure of Arabidopsis thaliana UDP-glucose 4-epimerase 2 (AtUGE2) complexed with UDP, wild-type |
| 52 | 8WTB | B | 187 | Crystal structure of McsA/McsB complex truncated by chymotrypsin |
| 53 | 8WU7 | A | 306 | Structure of a cis-Geranylfarnesyl Diphosphate Synthase from Streptomyces clavuligerus |
| 54 | 8X3S | B | 34 | Crystal structure of human WDR5 in complex with PTEN |
| 55 | 8XJE | B | 153 | Crystal structure of the YqeY protein from Campylobacter jejuni |
| 56 | 8XJG | A | 153 | Crystal structure of the YqeY protein from Vibrio parahaemolyticus |
| 57 | 8Y9P | A | 256 | Crystal structure of bacterial activating sulfotransferase SgdX2 |
| 58 | 8YXK | A | 201 | X-ray structure of Clostridioides difficile endolysin Ecd09610 glucosaminidase domain. |
| 59 | 9B1R | A | 562 | Functional implication of the homotrimeric multidomain vacuolar sorting receptor 1 from Arabidopsis thaliana |
| 60 | 9BCZ | A | 644 | Chicken 1-phosphatidylinositol 4,5-bisphosphate phosphodiesterase zeta-1 (PLCZ1) in complex with calcium and phosphorylated threonine |
| 61 | 9F63 | A | 572 | Crystal structure of Saccharomyces cerevisiae pH nine-sensitive protein 1 (PNS1) |

Table 5: Detailed information of enzyme data.

| ID | EC | Length | family |
|---|---|---|---|
| P40925 | 1.1.1.37 | 334 | malate dehydrogenase |
| Q7X7H9 | 2.7.1.71 | 287 | shikimate kinase |
| Q15165 | 3.1.1.2 | 354 | arylesterase |

introduced for our model to design protein sequence families separately and based on Multiple Sequence Alignments (MSAs), forming results by evaluation of CCMPRED (Seemayer et al., 2014). All targets were filtered from the CAMEO submitted target list, and those discovered before May 2024 were excluded to avoid potential data leakage. Three enzyme families were used to validate our model's ability to generate MSAs with correct functional annotations following (Song et al., 2024), with detailed information provided in Appendix E.1, forming result by a scoring model CLEAN (Yu et al., 2023). Additionally, lysozyme families were generated and folded into structures using ESM-Fold, following the PoET method (Truong Jr & Bepler, 2023) paper, thereby complementing our structural results. All detailed information about evaluation benchmarks are provided in D.2

## D EXPERIMENTAL DETAILS

### D.1 TRAINING CONFIGURATION

**Training Dataset** In line with ESM-2, we use protein sequence data from the UniRef database (Suzek et al., 2007) (as of March 2024) to train ProfileBFN. Our training data selection strategy also aligns with ESM-2, starting with an even selection of cluster groups from UniRef50 results, followed by random sequence selection within these clusters based on UniRef90 clustering. In total, the training involves 190 million protein sequences. Notably, although ProfileBFN utilizes MSA profiles as inputs, it does not require the construction of additional profile data, but merely uses existing sequence data for training, which greatly simplifies the implementation.

**Training Hyperparameters** We use the same Transformer (Vaswani, 2017) module as ESM-2 to implement ProfileBFN. For the ProfileBFN model with 650 million parameters, it has 33 layers of 20-head self-attention blocks. The hidden and embedding dimensions are 1280, and the feed-forward hidden size is 5120. Note that, unlike the ESM-2 model, we do not use any form of dropout for regularization, as the Bayesian flow itself provides sufficient stochasticity. For the Bayesian flow, $\beta(1)$ implies the uncertainty of the last step in the modeling procedure. Based on our empirical experience and cases in the original BFN paper (Graves et al., 2023), we found it could be approximately set according to the equation $beta(1) * K = \textbf{constant}$ ($K$ is the vocab size). With this principle, we could directly obtain a good setting of $\beta(1)$ following the previous empirical parameter in Graves et al. (2023) where $K$ is different. We consider three different candidate schedule functions for $\beta(t)$, linear, square and exponential, then we enumerate all three settings empirically over the small model (8M) and find linear works best in our task. We use AdamW (Loshchilov, 2017) to train our model, setting the learning rate at 0.0001, which linearly decays to a minimum of 4e-5. We adaptively set the batch size to approximately 2 million tokens.

### D.2 EVALUATION DETAILS

#### D.2.1 EVALUATION OF FAMILY PROTEIN GENERATION

**Settings** The evaluation for family protein generation involves multiple proteins as targets for generation, including 61 proteins from CAMEO (Robin et al., 2021), phage lysozyme proteins, and three enzyme proteins. Detailed information on these proteins can be found in the Appendix C.1. When using a profile as input, the hyperparameter $t_0$ is set to 0.6; when using a single sequence as input, it is set to 0.3. For the construction of the profile, we first perform an MSA search in the Uniclust30 database (Mirdita et al., 2017) using HHblits (Remmert et al., 2012) based on the natural sequence of the protein. Then, we obtain the profile according to the method described in the section 2.1. For each target protein, we require the model to generate 1000 sequences (without removing duplicates) for evaluation.

**Metrics** Since the goal of the family protein generation is to generate a cluster of diverse and novel proteins with similar structures and functions, our evaluation metrics are based on three dimensions: sequence, structure, and function.

For sequences, we expect the model to deliver diverse, and novel results. Therefore, we consider the diversity, and novelty of generated sequences as metrics.

- **Diversity:** A model that experiences mode collapse, where the generated outputs lack diversity and can only produce a limited number of different proteins, cannot provide users with a rich set of candidate results. We use the mean value of the identity between generated sequences as a metric to measure diversity, denoted as **Div.**

- **Novelty:** Similarly, a model that simply replicates the natural sequence is inadequate for supporting real-world design scenarios. A useful model needs to produce results that offer novelty. We measure novelty by calculating the maximum identity between the generated sequences and natural sequences, defined as $\frac{\sum_i (1 - \max_j(\text{identity}_{ij}))}{N}$, where $\text{identity}_{ij}$ denotes the identity between $i$th among $N$ generated sequence and $j$th reference sequence.

Proteins belonging to the same family typically exhibit high similarity in their tertiary structures. Therefore, structural evaluation of family protein generation primarily focuses on assessing whether the generated sequences contain the structural information corresponding to the proteins. For this purpose, we use the currently popular yet fragile parameterized instance-level evaluation metrics and more robust non-parametric cluster-level metrics for evaluation.

- **Parameterized instance-level:** Due to the promising advancements of protein structure prediction models such as AlphaFold2 (Jumper et al., 2021) and ESMFold (Lin et al., 2023), previous work has utilized these models to evaluate the structures of generated family sequences (Truong Jr & Bepler, 2023). Specifically, following Truong Jr & Bepler (2023), we use ESMFold to perform structure prediction for each generated family sequence and report the predicted local distance difference test value (**pLDDT**) output by ESMFold. Additionally, we compare the predicted structure with the natural reference structure and report the maximum template modeling score, denoted as **Max TM-score**.

- **Non-parametric cluster-level:** This metric is used to avoid incorrect model comparisons caused by bias in parameterized metrics. The instance-level metrics heavily rely on parameterized structure prediction models. However, Alkhouri et al. (2024) have pointed out that structure prediction models can also produce structures similar to natural proteins for adversarial samples based on the BLOSUM matrix (Henikoff & Henikoff, 1992). This undoubtedly undermines the reliability of parameterized metrics. Our experimental analysis shown in Appendix D.2.2 further illustrates that adversarial samples using the BLOSUM matrix merely replicate information contained in existing sequences, without providing new insights into our understanding of the family.

  Based on the observations above, we design a more robust non-parametric metric based on a cluster of sequences to avoid this issue. Specifically, we require the model to generate a cluster of sequences for a given family and explain the amino acid contacts in the reference structure by analyzing the mutations within the cluster using the non-parametric CCMpred tool (Seemayer et al., 2014). Following Lin et al. (2023), we report the precision of the top L (length of the protein), L/2, and L/5 predicted long-range contacts (amino acid sequence positions differ by 24 or more) as the corresponding metrics, denoted as **LR P@L**, **LR P@L/2**, and **LR P@L/5**. In addition, Long-range contacts are challenging to predict and are crucial for understanding protein structure, function, and valuable features (MacGowan et al., 2024).

The evaluation metrics for protein function are designed to assess whether newly generated protein members of a given family still retain similar functions. Strictly speaking, evaluating protein function requires wet lab experiments; however, this process is both expensive and time-consuming. Instead, we perform dry lab assessments based on a protein function classification model and have designed corresponding evaluation metrics. Specifically, we task the model with generating enzymes, a special type of protein, and classify the generated proteins using the widely adopted enzyme function classification model, CLEAN (Yu et al., 2023). We then assess whether the generated enzymes are correctly classified to determine if the family function is retained in the designs. The proportion of correctly classified results, after deduplication, is reported as a performance metric.

**Baselines** We select multiple strong protein design models as baseline models for comparison. Specifically, PoET (Truong Jr & Bepler, 2023) is an autoregressive model that uses known family sequences as prompts and generates new sequences for the family by continuously predicting the next amino acid. The model can generate sequences using either a single sequence or a multiple sequence alignment (MSA) as the prompt. Similar to us, EvoDiff (Alamdari et al., 2023) adopts a non-autoregressive generation approach. It leverages MSA to guide a discrete diffusion generation process, achieving the goal of family design. In a non-autoregressive paradigm, we have extended the powerful protein language model ESM-2 (Lin et al., 2023) to enable its application from protein understanding scenarios to family design. Specifically, for a given sequence in the family, we first mask 15% of the amino acids (consistent with the strategy used during training) and then iteratively replace the masked tokens with generated amino acids using ESM-2. The model most closely related to our ProfileBFN is DPLM (Wang et al., 2024). It is also trained on large-scale protein sequence datasets. However, while DPLM adopts a discrete diffusion framework, we utilize a Bayesian Flow Network capable of handling discrete data more smoothly. The original DPLM paper does not ad-

dress scenarios involving family designs. In this paper, we extend it to family designs by equipping it with a sampling strategy similar to ProfileBFN.

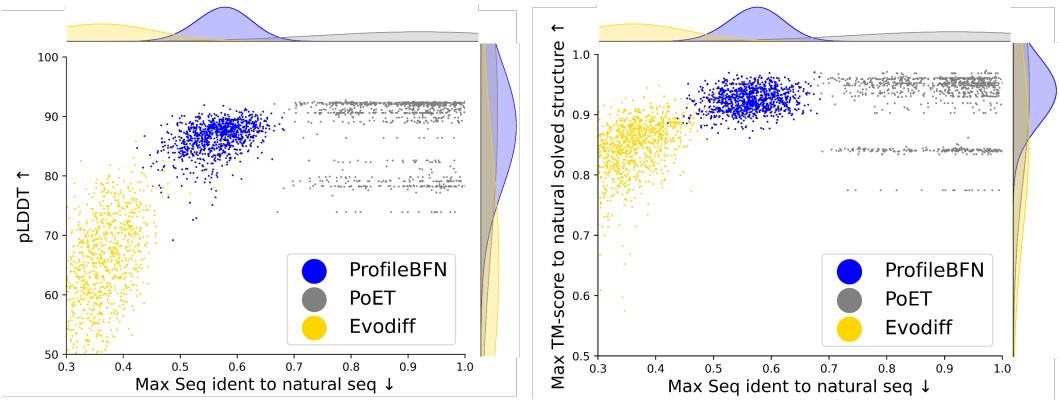

Figure 5: Sequence novelty and predicted structural conservation of phage lysozymes generated by ProfileBFN, PoET and EvoDiff. ProfileBFN effectively captures the conserved structural features of families while providing sufficient novelty.

### D.2.2 NON-PARAMETRIC: WHY IMPORTANT

We assert that non-parametric methods, like CCMPRED (Seemayer et al., 2014), offer distinct advantages in evaluating generated protein sequences. To validate our hypothesis, we have conducted additional BLOSUM62-based hacking experiments, which reveal how structural evaluations, such as ESMFold's pLDDT scores, may not perform optimally in certain respects.

To challenge the efficacy of ESMFold, we employed the BLOSUM62 (Henikoff & Henikoff, 1992) matrix to score the sequences after randomly substituting amino acid residues from the ground truth sequences. Subsequently, we selected those modified sequences with high scores and analyzed their predicted structures by ESMFold. With a sequence identity threshold set at 0.4, we observed that most of these hacked proteins still exhibited favorable pLDDT and pTM scores; however, their structures, as depicted in Figure 6, were erroneous and devoid of biological significance.

Additionally, we discovered that some protein samples generated by PoET faced a similar issue, indicating that ESMFold may not provide a comprehensive evaluation. As illustrated in Figure 7, sequences with repetitions and those following simple patterns still received high pLDDT scores from ESMFold. To some extent, the pLDDT scores in these cases reflect confidence because structures, such as those resembling a stick, are easily recognizable.

### D.2.3 EVALUATION OF PROTEIN REPRESENTATION LEARNING

**Settings** For the evaluation of protein representation learning, we assess the representations of ProfileBFN on various protein prediction tasks (Wang et al., 2024; Su et al., 2023; Dallago et al., 2021). These tasks include protein function prediction (Thermostability and Metal Ion Binding), protein localization prediction (DeepLoc), protein annotation prediction (EC and GO), and protein-protein interaction prediction (HumanPPI). Following Wang et al. (2024), we perform full-parameter supervised fine-tuning on each dataset.

**Metrics** We use accuracy (**ACC%**) as the primary evaluation metric for most tasks in representation learning since these tasks are primarily classification problems, **Accuracy** refers to the percentage of instances where the model accurately predicts the correct class for specific proteins in general it is computed as $\frac{\sum_1^N \mathbf{1}_{(y=\hat{y})}}{N}$, where $y$, $\hat{y}$ are the ground truth label and model predicted label, $N$ is the total number of samples. In the context of HumanPPI and Metal Ion Binding tasks, protein pairs are classified into two categories based on whether they interact. For the DeepLoc task, classifications are made either into 10 classes for subcellular localization or into 2 classes for binary localization.

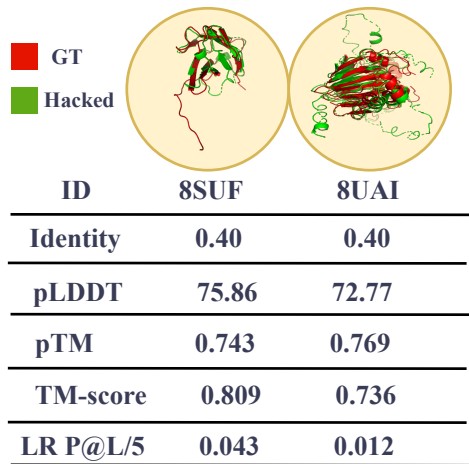

| ID | 8SUF | 8UAI |
|---|---|---|
| Identity | 0.40 | 0.40 |
| pLDDT | 75.86 | 72.77 |
| pTM | 0.743 | 0.769 |
| TM-score | 0.809 | 0.736 |
| LR P@L/5 | 0.043 | 0.012 |

Figure 6: Hacking ESMFold's pLDDT by BLOSUM62 Matrix

Spearman's rank correlation (Spearman's $\rho$) (Zar, 2005) coefficient is a statistical measure that evaluates the strength and direction of the association between two ranked variables. It quantifies the degree of monotonicity in the relationship, meaning it assesses how well the relationship between the two variables can be described by a monotonic function. In essence, it indicates whether an increase in one variable consistently corresponds to an increase or decrease in the other, regardless of whether the relationship is linear.

It is used to assess the relationship between the ground truth values of protein thermostability, as outlined by FLIP (Dallago et al., 2021), and the predicted values. Specifically, it is calculated as follows,

$$\rho = 1 - \frac{6 \sum d_i^2}{n(n^2 - 1)}, \quad d_i = \hat{x}_i - x_i \tag{34}$$

the prediction and ground truth are both ranked in descending order where $\hat{x}_i$ and $x_i$ indicates the predicted and ground truth rank.

Maximum F1-score (**Fmax**) is used for EC and GO annotation tasks. Fmax (Maximum F1-score) is a metric that balances the precision and recall of a classification model, reflecting the best trade-off between these two factors. In classification tasks, predictions can be categorized into four types: True Positive (TP), False Positive (FP), True Negative (TN), and False Negative (FN). A threshold $\lambda \in [0, 1]$ determines whether a prediction is considered True or False. Given $N$ model predicted scores $\{s_i \in [0, 1]\}_{i=1}^N$, corresponding labels are $\{l_i \in \{0, 1\}_{i=1}^N$, the True Positive (TP), False Positive (FP), True Negative (TN), and False Negative (FN) Precision (P), Recall (R), F1 score (F1)

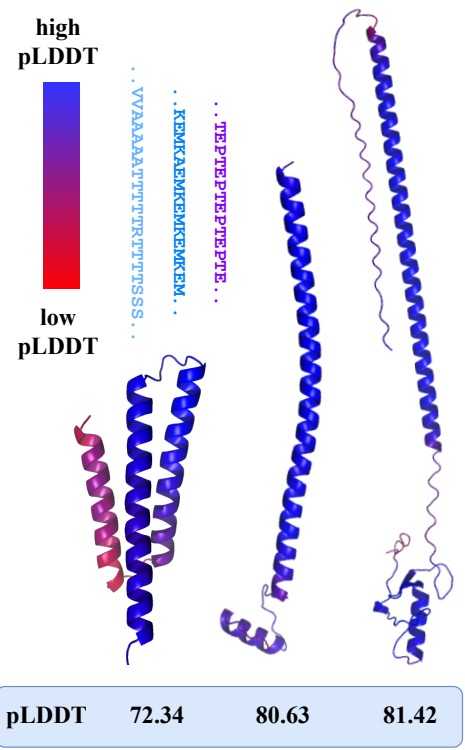

Figure 7: Trivial cases of PoET generated repeated sequence with high pLDDT after ESMFold.

and finally F$_{\text{max}}$ are subsequently calculated as follows:

$$N_{TP}(\lambda), \ N_{FP}(\lambda) = \sum_i l_i \mathbf{1}_{s_i \geq \lambda}, \ \sum_i l_i \mathbf{1}_{(s_i < \lambda)},$$

$$N_{TN}(\lambda), \ N_{FN}(\lambda) = \sum_i (1 - l_i) \mathbf{1}_{(s_i < \lambda)}, \ \sum_i (1 - l_i) \mathbf{1}_{(s_i \geq \lambda)},$$

$$P(\lambda) = \frac{N_{TP}(\lambda)}{N_{TP}(\lambda) + N_{FP}(\lambda)},$$

$$R(\lambda) = \frac{N_{TP}(\lambda)}{N_{TP}(\lambda) + N_{FN}(\lambda)},$$

$$F1(\lambda) = \frac{2P(\lambda)R(\lambda)}{P(\lambda) + R(\lambda)},$$

$$F_{\text{max}} = \max_\lambda \left( F1(\lambda) \right) \tag{35}$$

**Baselines** For evaluating representation learning, we use the following baselines: SaProt (Su et al., 2023) is a protein language model that is trained using sequence and structure tokens. MIF-ST (Yang et al., 2023) is a pre-training model that utilizes inverse folding structural guidance to enhance learning. ESM-1b (Rives et al., 2021) and ESM-2 (Lin et al., 2023) are two protein language models trained using masked language modeling. AR-LM (Wang et al., 2024) is a protein language model

trained based on autoregression. In contrast, DPLM (Wang et al., 2024) utilizes non-autoregressive discrete diffusion modeling and is the model most closely related to ProfileBFN.

# E    COMPLEMENTARY RESULTS

## E.1    ENZYME GENERATION

### E.1.1    BACKGROUND

Enzymes are a special class of proteins with catalytic functions. They significantly accelerate chemical reactions within organisms and play a crucial role in sustaining life processes. Based on the differences in the types of chemical reactions catalyzed by various enzyme families, researchers have developed the Enzyme Commission Number (EC Number) system to classify enzymes. In other words, two enzyme proteins sharing the same EC Number are considered to have similar catalytic functions. Strictly speaking, determining an enzyme's EC Number requires labor-intensive and costly wet-lab experiments. However, with advancements in machine learning, the accuracy of using computational methods to predict EC Numbers has improved significantly. Among these methods, CLEAN (Yu et al., 2023) is one of the most advanced models for predicting enzyme EC Numbers. It employs a contrastive learning strategy to bring representations of functionally similar enzymes closer while pushing dissimilar ones apart, achieving classification accuracy validated by wet-lab experiments.

### E.1.2    SETTINGS

Following the work of previous researchers, we selected three representative enzyme families for model evaluation following  (Song et al., 2024). These families possess distinct characteristics that make them important in biological research. Firstly, P40925, which belongs to the family of malate dehydrogenases, plays an essential role in the malate-aspartate shuttle and the tricarboxylic acid (TCA) cycle. It catalyzes the reduction of aromatic alpha-keto acids in the presence of nicotinamide adenine dinucleotide(NADH). Secondly, Q7X7H9, which belongs to the family of shikimate kinases, catalyzes the specific phosphorylation of the 3-hydroxyl group of shikimate acid. It is a key enzyme in the shikimate pathway, responsible for the biosynthesis of the aromatic amino acids phenylalanine, tyrosine, and tryptophan. Finally, Q15165 is capable of hydrolyzing lactones and a number of aromatic carboxylic acid esters. It possesses antioxidant properties, which are crucial in reducing intracellular and local oxidative stress and are related to the pathogenesis of various diseases. For each enzyme family, we require the model to generate 1,000 protein sequences for evaluation. For ProfileBFN, we convert the known protein sequences within the family into a profile, which serves as the input for generation. For each generated protein sequence, we use CLEAN to classify its function and verify whether it belongs to the given family.

### E.1.3    BASELINES

We have selected several models specialized in generating protein families for comparison. PoET (Truong Jr & Bepler, 2023) is an autoregressive model that uses known family sequences as prompts and generates new sequences for the family by continuously predicting the next amino acid. When generating new enzyme family sequences, known enzyme sequences are converted into prompts and input into PoET. PoET treats sequences of protein families as sequences-of-sequences, utilizing both attention modules to capture within-sequence and between-sequence relationships in a hierarchical manner. EvoDiff (Alamdari et al., 2023) employs a non-autoregressive generation approach, utilizing multiple sequence alignments (MSA) to guide a discrete diffusion generation process and achieve family-specific generation. For this method, known enzyme sequences within the family are organized into an MSA.

### E.1.4    METRICS

We used diversed benchmark to evaluate the performance of generated Enzyme sequences:

- **Accuracy:** Accuracy is defined as the percentage of sequence candidates classified by the CLEAN model into the correct class of EC numbering. Specifically, we deduplicate the sequences beforehand.

- **Uniqueness:** Considering that generative models may output the same sequences in different iterations, we record the survival rates before and after deduplication as Uniqueness.

- **A × U:** An aggregate indicator which is defined as **Accuracy × Uniqueness** to both measure model's ability to generate accurate and unique sequence.

Refer to Appendix. D.2.1 for **Novelty** and **Diversity** metric details.

### E.1.5 RESULTS

We present detailed experimental results in Table 5 and then present generated sequences it in 10, 11 and12.

Table 6: Additional results complementing Table 2 are provided to showcase our model's performance. Notably, our model achieves the highest Accuracy × Uniqueness. The MSA Depth indicates the depth of MSA that are used as input of the generation model

|  | Model | P40925 | Q7X7H9 | Q15165 |
|---|---|---|---|---|
| MSA Depth | - | 572 | 443 | 15 |
| Accuracy × Uniqueness ↑ | PoET | 3.00% | 33.3% | 0.05% |
|  | EvoDiff-MSA | 27.93% | 88.69% | 1.39% |
|  | ProfileBFN-profile | 95.19% | 98.98% | 42.67% |
| Accuracy ↑ | PoET | 98.04% | 99.93% | 100% |
|  | EvoDiff-MSA | 27.93% | 88.69% | 1.39% |
|  | ProfileBFN-profile | 95.19% | 98.98% | 42.67% |
| Uniqueness ↑ | PoET | 3.06% | 33.32% | 0.05% |
|  | EvoDiff-MSA | 100% | 100% | 100% |
|  | ProfileBFN-profile | 100% | 100% | 100% |
| Novelty ↑ | PoET | 0.036 | 0.366 | 0.068 |
|  | EvoDiff-MSA | 0.728 | 0.596 | 0.497 |
|  | ProfileBFN-profile | 0.467 | 0.582 | 0.288 |
| Diversity ↓ | PoET | 0.499 | 0.645 | 0.990 |
|  | EvoDiff-MSA | 0.138 | 0.184 | 0.143 |
|  | ProfileBFN-profile | 0.374 | 0.289 | 0.594 |

### E.2 IMPROVE STRUCTURE PREDICTION VIA ENHANCING MSA

### E.2.1 BACKGROUND

Orphan protein structure prediction is an important scientific challenge, aiming to improve the accuracy of models in predicting the structures of orphan proteins. Specifically, orphan proteins refer to those that lack sequence and structure homology information (Wu et al., 2022). Due to the absence of homologous data, it is difficult to construct high-quality Multiple Sequence Alignments (MSAs) for these proteins (Chen et al., 2024). The low quality of MSAs strongly limits the performance of current structure prediction models, such as the AlphaFold series (Wu et al., 2022) (Chen et al., 2024). Moreover, orphan proteins are not uncommon in the protein space; statistics show that approximately 20% of metagenomic proteins and around 11% of proteins from eukaryotic and viral origins are classified as orphan proteins (Chen et al., 2024). Therefore, addressing orphan protein structure prediction remains a critical challenge in the post-AlphaFold era.

### E.2.2 BASELINES

The current advanced approach to addressing this issue is to use generative models to enhance low-quality MSAs, transforming them into high-quality MSAs. Based on this paradigm, MSAGPT(Chen

et al., 2024) reports the best predictive performance to date. Specifically, MSAGPT employs an autoregressive model, taking low-quality MSAs as input and sampling additional protein sequences to improve the quality of the MSAs. MSAGPT outperforms several methods that enhance MSAs to boost predictive performance, including EvoDiff (Alamdari et al., 2023), MSA-Aug(Zhang et al., 2023b), and EvoGen(Zhang et al., 2023a). Due to its advanced performance, we use MSAGPT as the main baseline method. Additionally, we treat the performance of AlphaFold2 using non-enhanced MSAs on orphan proteins as a lower bound, referred to as AF2-MSA. It is worth noting that while ProfileBFN, like MSAGPT, also enhances MSAs using generative models, it differs from MSAGPT in that its training only requires protein sequence data, which is more easily accessible. In contrast, MSAGPT requires training on MSA datasets and is further optimized based on AlphaFold2 feedback using Reinforcement Learning.

### E.2.3 SETTINGS

We follow MSAGPT and evaluate the model using orphan proteins from the CASP14 and CASP15 datasets. For each orphan protein, we retrieve its MSA using HHblits(Remmert et al., 2012) from the UniClust30 database(Mirdita et al., 2017). The obtained MSA has a depth of less than 20, meaning fewer than 20 homologous sequences can be retrieved. Generation models are required to generate 64 additional protein sequences based on the retrieved low-quality MSA. These sequences supplement the retrieved MSA, forming a higher-quality MSA. This high-quality MSA is then used as input for AlphaFold2 to improve its structural prediction performance for orphan proteins. In utilizing the retrieved MSA, ProfileBFN transforms it into a profile to be used as model input, while MSAGPT uses it as a prompt to guide the model in generation.

### E.2.4 METRICS

We compare the performance of different methods by analyzing the differences between the orphan protein structures predicted by AlphaFold2 and those obtained experimentally. Specifically, we use two golden metrics: TM-score, a widely-used metric for assessing the structural similarity between predicted structures and the ground truth, and LDDT, the Local Distance Difference Test score, which measures how well local interactions in a reference structure are conserved in the protein model being assessed. Additionally, we report a predictive metric, pLDDT (predicted Local Distance Difference Test), which reflects AlphaFold2's confidence in the local accuracy of each residue. All metrics are scaled from 0 to 100.

### E.2.5 RESULTS

Table 7 presents the performance metrics of different methods. Based on this table, we observe the following findings:

- Generating additional protein sequences can indeed enhance the quality of MSA, thereby improving the model's performance. This improvement stems from the model's pretraining process, which enables it to gain a profound understanding of protein structures. When applied to orphan proteins, the model effectively transfers this understanding, enriching initially low-quality MSAs with structural information and ultimately yielding high-quality MSAs.

- ProfileBFN consistently outperforms MSAGPT across all metrics, demonstrating that the MSA supplements provided by ProfileBFN contain more comprehensive protein structure information. This result can be attributed to several factors. First, ProfileBFN leverages pretraining to capture deeper protein structural insights compared to MSAGPT, as its non-autoregressive strategy aligns more closely with the natural characteristics of protein data. Second, the structural information obtained by ProfileBFN is more transferable to orphan proteins, unlike MSAGPT, whose pretraining primarily relies on deeper MSAs, while ProfileBFN imposes no specific depth requirement on MSAs.

Table 7: Using ProfileBFN to enhance AF2 performance by adding virtual MSAs, the results show that ProfileBFN is capable of generating more appropriate MSAs for models such as AF2 compared to the ground truth searched MSA and MSAGPT. All metrics are scaled from 0 to 100.

| Model | TMscore ↑ | LDDT ↑ | pLDDT ↑ |
|---|---|---|---|
| AF2-MSA | 53.20 | 54.01 | 62.91 |
| MSAGPT | 55.72 | 55.59 | 66.38 |
| ProfileBFN | **56.84** | **55.72** | **67.04** |

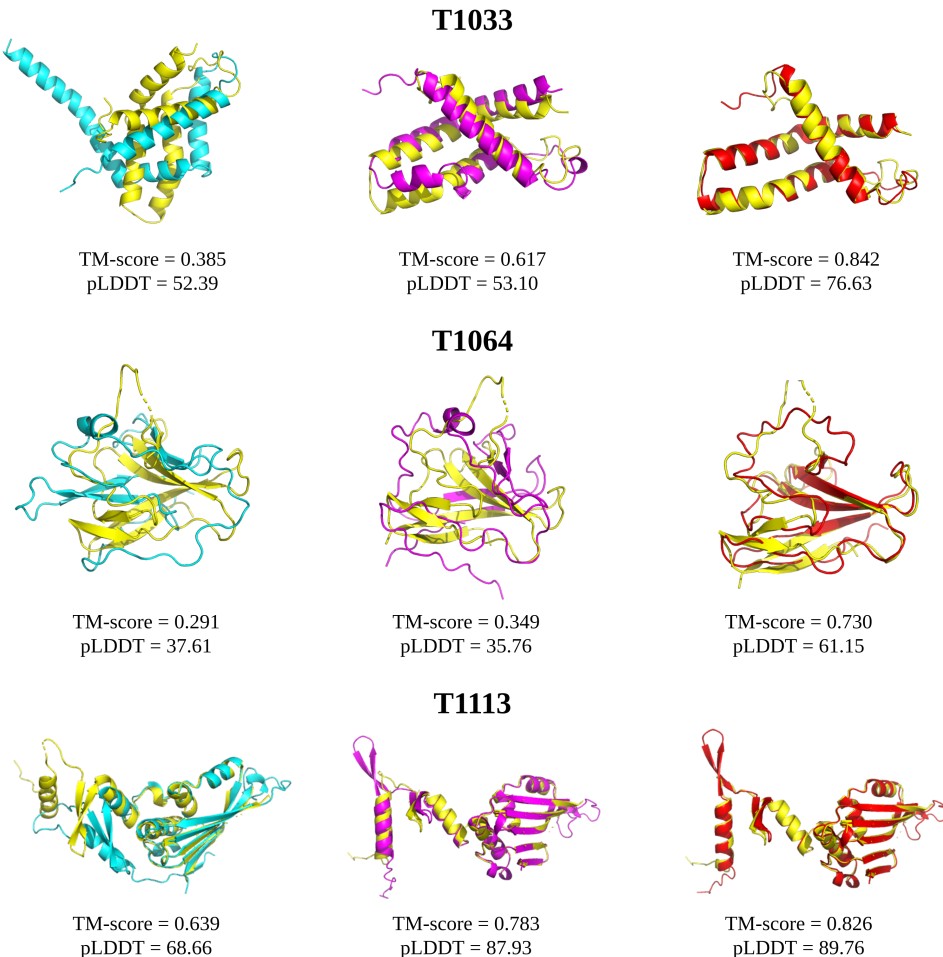

Figure 8: Visualization of improved structure prediction sample compared with AlphaFold2 and MSAGPT. Yellow: Ground truth; Blue: Predictions from MSA generated by natural MSA searched with AF2. Purple: Predictions based on MSA generated by MSAGPT; Red: Predictions based on MSA generated by ProfileBFN;

### E.3 ANTIBODY CDR IN-PAINTING

### E.3.1 SETTINGS

We further test our Model's ability in the task of Antibody Complementarity Determining Regions (CDR) in-painting. Antibodies are specific types of proteins utilized by the immune system to recognize and neutralize pathogens and are of immense interest for therapeutics. In the structure of antibodies, the so-called Complementary-Determining Regions are the main regions for binding

with antigens and determining the specificity of the antibodies. Under this circumstance, CDRs in antibody sequences are masked at once and later predicted conditioned on the framework. We present two versions of our model for antibody generation: ProfileBFN-single(650M) without any information trained about antibodies, and ProfileBFN-Anti(650M), which is tuned with the OAS dataset for 8500 steps.

### E.3.2  BASELINES

We include several strong baselines, all of which are trained specifically on antibody data. RAbD (Adolf-Bryfogle et al., 2018) is a renowned software-based method. DiffAb (Luo et al., 2022) uses diffusion models to conduct sequence-structure co-design, which mainly models the geometric aspect. AntiBERTy (Ruffolo et al., 2021) and AbLang (Olsen et al., 2022) are two sequence-based language models trained on the entire OAS dataset; the former is based on the BERT architecture to encode antibody sequences, while the latter is trained on randomly masked antibody sequences and modeled on a Transformer architecture with a special head.

### E.3.3  METRICS

We used Amino Acid recovery (AAR) of each CDR region for evaluation, with each antibody sample providing 5 candidates.

### E.3.4  DATASETS

We used the OAS unpaired dataset (total 2,428,016,345 antibody sequences) to fine-tune our model and the SAbDab Dataset for testing, following the DiffAb paper. To avoid potential data leakage, we removed sequences similar to our test set with MMSeqs2 (Steinegger & Söding, 2017) tools by the identity of 0.95. Both heavy chains and light chains are included in the tuning process.

### E.3.5  RESULTS

The results showed in Table E.3.5 indicate that ProfileBFN had already reached comparable scores before fine-tuning on the antibody dataset, indicating that it learned general rules of protein language that could be successfully transferred to antibodies which are specific and functional proteins. Once tuned on the antibody dataset for a very small number of steps, it could surpass the performance of previous models such as AntiBERTy and AbLang, indicating the effectiveness of pre-training processes.

| Model | CDR-H1 | CDR-H2 | CDR-H3 | CDR-L1 | CDR-L2 | CDR-L3 |
|---|---|---|---|---|---|---|
| RAbD | 0.2285 | 0.2550 | 0.2214 | 0.3427 | 0.2630 | 0.2073 |
| DiffAb | 0.6575 | 0.4931 | 0.2678 | 0.5667 | 0.5932 | 0.4647 |
| AntiBERTy | 0.7940 | 0.5932 | **0.4133** | **0.7208** | 0.3996 | 0.2758 |
| AbLang | 0.7039 | **0.7981** | 0.3207 | 0.5799 | 0.5513 | 0.3175 |
| ProfileBFN-single | 0.6766 | 0.6188 | 0.1946 | 0.5356 | 0.5873 | 0.3064 |
| ProfileBFN-Anti | **0.8227** | 0.7236 | 0.3343 | 0.6402 | **0.6156** | **0.4716** |

Table 8: Performance of Antibody CDR in-paint task ProfileBFN compared to baselines. The best result is indicated in bold, while the second-best result is underlined.

### E.4  ADDITIONAL RESULTS

#### E.4.1  INVESTIGATION ON THE RELATIONSHIP BETWEEN PERFORMANCE AND MSA DEPTH

We have conducted the experiment on a case, where we sampled 50, 100, 500, 1000, and 2000 sequences from the searched homologous sequences, and each generate 1000 sequences for contact prediction, we report LR P@L, LR P@L/2, LR P@L/5 respectively. Results shown in Fig. 9 reveal that the quality of generated sequences tends to increase with the increasing depth of the MSA. The growth rate drops as the depth increases.

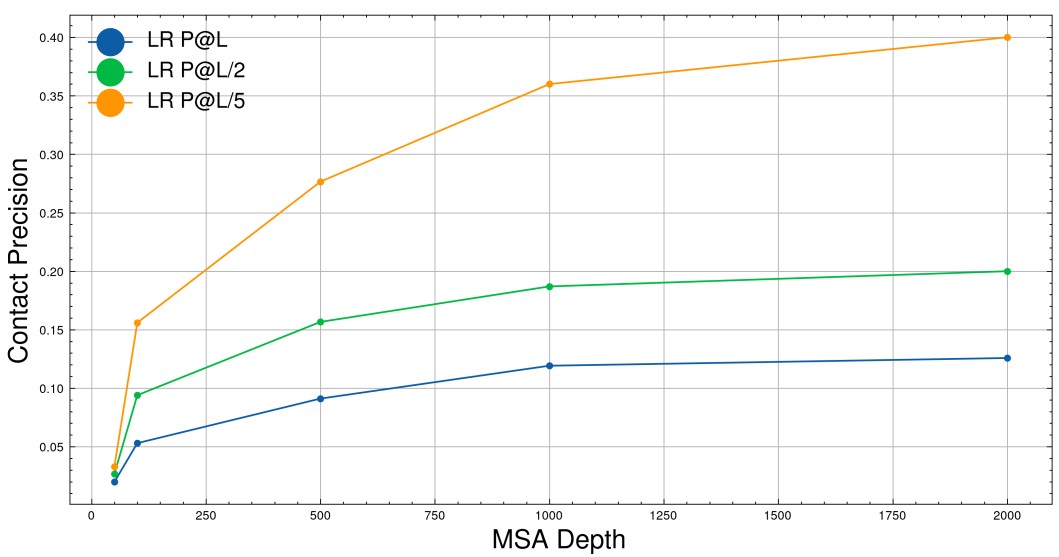

Figure 9: ProfileBFN-profile Generation Result of Contact Prediction of Protein 8YXK with different MSA depth as input

Figure 10: Samples of sequences conditioned on enzyme P40925 family by model ProfileBFN, PoET and EvoDiff;

Figure 11: Samples of sequences conditioned on enzyme Q7X7H9 family by model ProfileBFN, PoET and EvoDiff;

Figure 12: Samples of sequences conditioned on enzyme Q15165 family by model ProfileBFN, PoET and EvoDiff;

