# OpenReview forum: "Steering Protein Family Design through Profile Bayesian Flow"
_ICLR.cc/2025/Conference — ICLR 2025 Oral_

### Official Review · Reviewer_vGzf · 2024-10-29

**Soundness:** 3
**Presentation:** 3
**Contribution:** 4
**Rating:** 8
**Confidence:** 3

**Summary:**

The authors introduce Profile Bayesian Flow Networks (ProfileBFN), an adaptation of the Bayesian Flow Networks from Graves et al. (2023), designed specifically for aligned protein sequences in multiple sequence alignments (MSAs) and their frequency profiles. They derive a simplified loss function measuring the squared difference between the predicted latent embedding p_phi and the encoded representation e_x, effectively capturing how well the model’s predictions align with encoded information in the data.  Furthermore, the authors present techniques to efficiently sample from the model.
Arguing that training directly on MSA profiles rather than raw MSAs offers key advantages, the authors highlight that MSAs can be large, computationally intensive, and challenging to generate. By relying on sequence profiles instead, their method reduces computation and even enables training on single sequences rather than full MSAs. Furthermore, the approach does not require additional profile data construction, but instead leverages existing sequence profile data for training.
Their results show that the generated sequences better capture protein contacts than standard MSA-based methods. Additionally, in-silico refolding of these generated sequences predicts structures that closely resemble native structures. Despite these advances, the generated sequences remain highly similar to native sequences in terms of sequence identity.

**Strengths:**

The papers' emphasis is focused on the mathematical derivation and has clear explanations of key theoretical components. The authors benchmark ProfileBFN against multiple SOTA methods, demonstrating that it performs on par with or surpasses existing approaches.

**Weaknesses:**

While the paper is strong on the mathematical side, it lacks some depth in application.

**Questions:**

To better showcase the strength and applicability of ProfileBFN, I’d suggest the following points:
* The authors claim that their method is suitable for de-novo design, potentially enabling the creation of novel sequences, structures, or even entirely new functions. However, it appears that the sequences generated by ProfileBFN are optimized variants closely resembling native sequences in terms of similarity and identity. Could the authors elaborate on this point and clarify how ProfileBFN supports de-novo design?
* Could the authors specify which protein families were used to evaluate their method? In particular, is there a dependency between the depth of the MSA and the quality of the generated sequences? Certain regions, such as antibody loops (and loops in general), are challenging to model and generate due to the complexity of viable variations. Have the authors considered testing ProfileBFN in these difficult sequence spaces?

---

> ### Author Response · Authors · 2024-11-20
> **Response to Reviewer vGzf (1/3)**
>
> Thank you for your positive feedback, we addressed the questions raised by the reviewer in the following:
>
> **Q1: The authors claim that their method is suitable for de-novo design, potentially enabling the creation of novel sequences, structures, or even entirely new functions. However, it appears that the sequences generated by ProfileBFN are optimized variants closely resembling native sequences in terms of similarity and identity. Could the authors elaborate on this point and clarify how ProfileBFN supports de-novo design?**
>
> Our primary focus is on the family protein design, which generates protein sequence families conditioned on a single sequence or multiple sequence alignments (MSAs).
>
> In response to the reviewer's question, we have adapted our model to the task of unconditional generation, and we have received visually good samples as shown in **Fig. 9** in our revised paper. We initiate sequences randomly and set the initial time $t_0$ to 0, sequences are generated and then folded by ESMFold. The colors in Fig. 9, ranging from red to blue, represent lower to higher pLDDTs predicted by ESMFold.
>
> **Q2: Could the authors specify which protein families were used to evaluate their method?**
>
> There are 3 datasets for ProfileBFN's evaluation:
>
> 1. The dataset derived from CAMEO: 61 Multiple Sequence Alignments We collected 61 primary sequences from CAMEO dated from May 4, 2024, for each of the primary sequences, we searched for their homologous sequences using the same procedure as described in AlphaFold2.
>
> 2. *Three Enzyme families* are chosen from EnzyGen [3], three representative categories of catalytic enzymes, all of which are extensively validated experimentally:
>     - *EC 1.1.1.37* represented by P40925: The family of malate dehydrogenase, which plays an essential role in the malate-aspartate shuttle and the tricarboxylic acid (TCA) cycle, catalyzing the reduction of aromatic alpha-keto acids in the presence of nicotinamide adenine dinucleotide (NADH).
>     - *EC 2.7.1.71* represented by Q7X7H9: The family of shikimate kinases, which are key enzymes in the shikimate pathway, responsible for the biosynthesis of the aromatic amino acids phenylalanine, tyrosine, and tryptophan, catalyzing the specific phosphorylation of the 3-hydroxyl group of shikimate acid.
>     - *EC 3.1.1.2* represented by Q15165: The family of arylesterases, which possess antioxidant properties and are crucial in reducing intracellular and local oxidative stress, and are related to the pathogenesis of various diseases.
>
> 3. *Phage lysozyme families* represented by Q37875: The family of lysozyme following PoET [4] and ProGen [5] which plays a crucial role in the life cycle of bacteriophages, capable for degrading bacterial cell walls.
>
> We provide additional information regarding these items in **Table 4**, **Table 5** in our revised paper.
>
>
> **Q3：In particular, is there a dependency between the depth of the MSA and the quality of the generated sequences?**
>
> Thanks for the constructive question, we have conducted new experiments in response, and the reuslt are presented as **Figure 10** in the revised paper.
>
> The resuslts reveals that the quality of generated sequences tends to increase with the increasing depth of the MSA. the growth rate drops as the depth increase.
>
> Specifically, we conducted the experiment on a case, where we sampled 50, 100, 500, 1000, 2000 sequence from the searched homologous sequences, and each generate 1000 sequences for contact prediction, we report LR P@L, LR P@L/2, LR P@L/5 respectively.

---

> ### Author Response · Authors · 2024-11-20
> **Response to Reviewer vGzf (2/3)**
>
> **Q4: Certain regions, such as antibody loops (and loops in general), are challenging to model and generate due to the complexity of viable variations. Have the authors considered testing ProfileBFN in these difficult sequence spaces?**
>
> Thank you for the advice. We have evaluated ProfileBFN's performance on the task of Antibody CDR in-painting. To address this task, we masked all CDR regions of the antibody simultaneously and predicted them based on the remaining framework and use our proposed model to do the in-painting.  The results are shown in the table below:
>
> | Model  | CDR-H1     | CDR-H2     | CDR-H3     | CDR-L1     | CDR-L2     | CDR-L3     |
> | ----------------------- | ---------- | ---------- | ---------- | ---------- | ---------- | ---------- |
> | RAbD                    | 0.2285     | 0.2550     | 0.2214     | 0.3427     | 0.2630     | 0.2073     |
> | DiffAb                  | 0.6575     | 0.4931     | 0.2678     | 0.5667     | 0.5932     | 0.4647     |
> | AntiBERTy               | 0.7940     | 0.5932     | **0.4133** | **0.7208** | 0.3996     | 0.2758     |
> | AbLang                  | 0.7039     | **0.7981** | 0.3207     | 0.5799     | 0.5513     | 0.3175     |
> | **ProfileBFN-single** | 0.6766     | 0.6188     | 0.1946     | 0.5356     | 0.5873     | 0.3064     |
> | **ProfileBFN-Anti** | **0.8227** | 0.7236 | 0.3343     | 0.6402     | **0.6156** | **0.4716** |
>
> Where, all numbers reported are Amino Acid recovery (AAR). ProfileBFN-single represents our model performing the task without tuning on the domain. Meanwhile, ProfileBFN-Anti was fine-tuned on the OAS-unpaired dataset for 8500 steps, focusing on both the heavy and light chains. For testing, we employed the SAbDab test set, consistent with DiffAb's framework.
>
> We included several strong baselines, RAbD [6] and DiffAb [7] for Sequence and Structure co-design, AntiBERTy [8] and AbLang [9] for language modeling, all of which were trained on antibody dataset.
>
> From the table above, we can draw two conclusions:
>
> 1. **ProfileBFN proved its effectiveness by surpassing most of the baseline models**, achieving 3 best results and 3 second-best results out of a total of 6 CDR regions.
>
> 2. **ProfileBFN captures inherent protein rules**
> The fact that ProfileBFN-single is able to reach a comparable performance without training on data in Antibody domain, shows that ProfileBFN have captured inherent interactions.
>
> Detailed information about the entire experiment with antibodies can be found in **Appendix E.4** in our revised paper.

---

> ### Author Response · Authors · 2024-11-20
> **Response to Reviewer vGzf (3/3)**
>
> **Q5: The paper lacks some depth in application**
>
> We prove ProfileBFN's perspect in application by utilizing it to solving the problem of Orphan protein structure prediction, where vitual MSAs are generated to enhance the accuracy of AlphaFold2 structure prediction. Results are as follows:
> | Model     | TMscore&uarr; | LDDT&uarr;      | pLDDT&uarr;     |
> | --------- | --------- | --------- | --------- |
> | AF2-MSA   | 53.20     | 54.01     | 62.91     |
> | MSAGPT    | 55.72     | 55.59     | 66.38     |
> | **ProfileBFN** | **56.84** | **55.72** | **67.04** |
>
> Orphan proteins refer to those that lack sequence and structure homology information and low-quality Multiple Sequence Alignments (MSAs). Thus it limits the performance of current structure prediction models, such as the AlphaFold series. Moreover, orphan proteins stands for approximately 20% of metagenomic proteins and around 11% of proteins from eukaryotic [10]. We would like to generate High-quelity MSAs based on Currently Low-quelity MSAs, which improve the performance of AlphaFold2 and is valuable to solve the issue of orphan sequences.
>
> We import MSAGPT[10], a model employs an autoregressive model and trained on MSA data as well as AlphaFold2 feedback through Reinforcement Learning, as our main baseline as it reports the best predictive performance to date on this task. AF2-MSA stands for AlphaFold2 using non-enhanced MSAs on orphan proteins as a lower bound. TM-score, LDDT and pLDDT are reported after folded with AlphaFold2 conditioned on High-quelity MSAs. Noted that pLDDT only reflects AlphaFold2's confidence in the local accuracy of each residue. All metrics are scaled from 0 to 100.
>
> From the results above we make assertions that ProfileBFN could best enhance AlphaFold2's performance by generating additional protein sequences without any additional training with MSA and AlphaFold2 feedback.This shed light on ProfileBFN's potiential application on Protein Structure prediction.
>
> **Reference:**
>
> [1] Xavier Robin, Juergen Haas, Rafal Gumienny, Anna Smolinski, Gerardo Tauriello, and Torsten Schwede. Continuous automated model evaluation (cameo)—perspectives on the future of fully automated evaluation of structure prediction methods. Proteins: Structure, Function, and Bioin-formatics, 89(12):1977–1986, 2021.
>
> [2] Stefan Seemayer, Markus Gruber, and Johannes S¨oding. Ccmpred—fast and precise prediction of protein residue–residue contacts from correlated mutations. Bioinformatics, 30(21):3128–3130, 2014.
>
> [3] Zhenqiao Song, Yunlong Zhao, Wenxian Shi, Wengong Jin, Yang Yang, and Lei Li. Generative enzyme design guided by functionally important sites and small-molecule substrates. arXiv preprint arXiv:2405.08205, 2024.
>
> [4] Timothy Truong Jr and Tristan Bepler. Poet: A generative model of protein families as sequences-of-sequences. Advances in Neural Information Processing Systems, 36:77379–77415, 2023.
>
> [5] Ali Madani, Bryan McCann, Nikhil Naik, Nitish Shirish Keskar, Namrata Anand, Raphael R Eguchi, Po-Ssu Huang, and Richard Socher. Progen: Language modeling for protein generation. arXiv preprint arXiv:2004.03497, 2020.
>
> [6] Jared Adolf-Bryfogle, Oleks Kalyuzhniy, Michael Kubitz, Brian D Weitzner, Xiaozhen Hu, Yumiko Adachi, William R Schief, and Roland L Dunbrack Jr. Rosettaantibodydesign (rabd): A general framework for computational antibody design. PLoS computational biology, 14(4):e1006112, 2018.
>
> [7] Shitong Luo, Yufeng Su, Xingang Peng, Sheng Wang, Jian Peng, and Jianzhu Ma. Antigen-specific antibody design and optimization with diffusion-based generative models for protein structures. Advances in Neural Information Processing Systems, 35:9754–9767, 2022.
>
> [8] Jeffrey A Ruffolo, Jeffrey J Gray, and Jeremias Sulam. Deciphering antibody affinity maturation
> with language models and weakly supervised learning. arXiv preprint arXiv:2112.07782, 2021.
>
> [9] Tobias H Olsen, Iain H Moal, and Charlotte M Deane. Ablang: an antibody language model for completing antibody sequences. Bioinformatics Advances, 2(1):vbac046, 2022.
>
> [10] Bo Chen, Zhilei Bei, Xingyi Cheng, Pan Li, Jie Tang, and Le Song. Msagpt: Neural prompting protein structure prediction via msa generative pre-training. arXiv preprint arXiv:2406.05347, 2024.

---

> > ### Comment · Reviewer_vGzf · 2024-11-25
> >
> > Thank you for addressing my concerns, clarifying your explanations, and incorporating additional experiments—especially the antibody loop one. I have increased my score for this work.

---

### Official Review · Reviewer_BQXP · 2024-11-02

**Soundness:** 3
**Presentation:** 2
**Contribution:** 4
**Rating:** 8
**Confidence:** 3

**Summary:**

Introduces a novel modification to the pre-existing Bayesian Flow Networks (BFN) learning scheme to create the ProfileBFN. It then applies this new ProfileBFN to the protein generation task. Like AlphaFold (and many other works) they use MSA, however, they use the the distribution statistics of MSA rather than the actual MSA data directly. They show that they technique performs well across most metrics, and produces state of the art results in some.

**Strengths:**

The results are overall quite impressive. Despite being outperformed in some metrics, ProfileBSN still achieves decent results in those metrics while achieving state of the art results in other measurements.

Additionally, I like the use of Bayesian Flow Networks, which are not entirely in the mainstream techniques in protein modeling. This gives this paper a high degree of novelty

I appreciate the introduction to BFNs provided in Section 2.2. This greatly helped the readability.

**Weaknesses:**

The mathematics is not presented in the easiest way to understand. It is not so problematic that I reject the submission, but it is not written in an easy manner for the reader to follow. Reuse of variable names (see Questions) is an issue, but the largest issues are:

A - In line 170: dimensions are given for the summation symbol. This does not make sense, as summation is an operator, not a data type. Please explain what this means. I have held off on lowering the score too much for it in hopes there is a good explanation, but I definitely would like to see an answer for this.

B - Explanation for many of the results feels quite rushed, and I had to put a lot of effort into parsing it out.

**Questions:**

1 - Why do you use sigma for both the Kronecker delta and the Dirac delta functions? This is confusing.

2 - You do not define PMF. What does it stand for?

3 - You do not define phi, it just appears in equation 3 (and in the line above). What does phi represent?

4 - Why do you not cite the paper (Z Lin’s or another) that introduces ESM-2?

5 - How did you choose the hyper parameters in lines 266-267?

---

> ### Author Response · Authors · 2024-11-20
> **Response to Reviewer BQXP (1/2)**
>
> Thank you for your positive feedback and valuable suggestions, which will help us further refine and enhance our work. We address the concerns raised by the reviewer as follows:
>
> **W1: In line 170: dimensions are given for the summation symbol. This does not make sense, as summation is an operator, not a data type. Please explain what this means. I have held off on lowering the score too much for it in hopes there is a good explanation, but I definitely would like to see an answer for this.**
>
> Sorry for the confusion, the symbol `\bm{\Sigma}`($\mathbf{\Sigma}$) in that line was intended to represent the covariance matrix of a multivariate Gaussian distribution which resembles a summation operator `\sum` under the render of Latex.
>
> We understand that this usage could be confusing, therefore to rectify this potential misunderstanding, we have replaced it with a new symbol `\mathcal{C}`($\mathcal{C}$) for the representation of a covariance matrix, throughout the paper.
>
> For further details on this revision, please refer to line 138-139, and line 175-176 in the updated version of our paper. We have made these changes in hopes of improving the overall quality and clarity of our work. We appreciate your feedback and are committed to ensuring that our paper is as accurate and understandable as possible.
>
> **W2: Explanation for many of the results feels quite rushed, and I had to put a lot of effort into parsing it out.**
>
> Apologies for any confusion this may have caused. We have thoroughly revised a significant portion of our work, including mathematical and experimental details both in the main part and appendix. The changes made are as follows:
>
> - The explanation for **Table 1** can be found line 268-303 and more detailed in **Appendix D.2.1**, including how the dataset was constructed, how family protein sequences are sampled from the model and how all metrics are calculated.
> - The explanation for **Table 2**, regarding the generation of functional enzymes can be found in line 365-383, details have been added to **Appendix E.1**. This includes how these enzymes were selected, how the virtual enzyme family MSA was sampled, and how our metrics were calculated. Furthermore, we have introduced **Table 6** to provide further details on **Table 2**.
> - The explanation for **Table 3** has been incorporated into line 390-398 and **Appendix D.2.3** including the definition and calculation method for Fmax and Spearman's correlation.

---

> > ### Comment · Reviewer_BQXP · 2024-11-21
> > **Response to rebuttal 1/2**
> >
> > Thank you for the updates.
> > W1: The symbol change is clear to me now, I have no further concerns over this.
> > W2: I appreciate the extensive re-working of the explanation. My main concern was over a reader's ability to understand the paper without referencing the appendix, and I am satisfied with the updates made to the main text. Additionally, the updates made to the appendix (especially D2.3, E1) provide substantial information for an interested reader.
> >
> > I have no further concerns regarding W1, W2.

---

> ### Author Response · Authors · 2024-11-20
> **Response to Reviewer BQXP (2/2)**
>
> **Q1: Why do you use sigma for both the Kronecker delta and the Dirac delta functions? This is confusing.**
>
> Thanks for pointing out the symbol abuse, the Kronecker delta function, which was previously denoted by sigma, has now been changed to `\bm{1}_{(\cdot = \cdot)}`. This change ensures that it is rendered as $\mathbf{1}_{(\cdot = \cdot)}$, adhering to the standard guideline representation for the Kronecker delta.
>
>
> **Q2: You do not define PMF. What does it stand for?**
>
> Sorry for the confusion, PMF stands for "Probability Mass Function," which has been updated in line 179.
>
> >The probability mass function specifies the probability that a discrete random variable is exactly equal to a particular value.
>
> In our revised version of the paper, we have made sure to expand all abbreviations the first time they appear.
>
> **Q3: You do not define phi, it just appears in equation 3 (and in the line above). What does phi represent?**
>
> The symbol $\mathbf{\phi}$ is defined to be the parameter of the neural network. We have revised the paper by adding clarification to the main text in line 122-123
>
> **Q4: Why do you not cite the paper (Z Lin’s or another) that introduces ESM-2?**
>
> We appreciate the reviewer's suggestion. In response, we have now included citations to Z Lin's paper on ESM series [1] [2] and [3]. We acknowledge the significance of ESM-2 as a key baseline in our research and have ensured that our work properly recognizes its contributions to the field.
>
>
> **Q5: How did you choose the hyperparameters in lines 266-267?**
>
> Thanks for pointing it out. The general hyperparameter, e.g. learning rate/batch sizes, is selected by grid-search over a small-scale model (8M). We delve into the two specific hyperparameters: $\beta(1)$ implies the uncertainty of the last step in the modeling procedure. Based on our empirical experience and cases in the original BFN paper[4], we found it could be approximately set according to the equation $beta(1) * K = \textbf{constant}$ ($K$ is the vocab size).  With this principle, we could directly obtain a good setting of  $\beta(1)$  following the previous empirical parameter in [4] where $K$ is different; We consider three different candidate schedule functions for $\beta(t)$, i.e. linear, square and exponential. And we enumerate all three settings empirically over the small model and find linear works best in our task. The above discussion has been updated in the revised paper.
>
> [1] Zeming Lin, Halil Akin, Roshan Rao, Brian Hie, Zhongkai Zhu, Wenting Lu, Nikita Smetanin, Allan dos Santos Costa, Maryam Fazel-Zarandi, Tom Sercu, Sal Candido, et al. "Language models of protein sequences at the scale of evolution enable accurate structure prediction"
>
> [2] Zeming Lin, Halil Akin, Roshan Rao, Brian Hie, Zhongkai Zhu, Wenting Lu, Nikita Smetanin, Robert Verkuil, Ori Kabeli, Yaniv Shmueli, et al. "Eevolutionary-scale prediction of atomic-level protein structure with a language model."
>
> [3] Tomas Hayes, Roshan Rao, Halil Akin, Nicholas J Sofroniew, Deniz Oktay,  Zeming Lin, Robert Verkuil, Vincent Q Tran, Jonathan Deaton, Marius Wiggert, and others. "Simulating 500 million years of evolution with a language model"
>
> [4] Graves, Alex and Srivastava, Rupesh Kumar and Atkinson, Timothy and Gomez, Faustino. "Bayesian flow networks"

---

> > ### Comment · Reviewer_BQXP · 2024-11-21
> > **Response to rebuttal 2/2**
> >
> > Q1: Thank you for the change in notation. I think the symbols are clear and match standard practice.
> > Q2: Thank you for the explanation. While PMF is fairly well known, I am always concerned about readers for which this paper resides on the periphery of their knowledge base.
> > Q3: Thank you for the clarity update.
> > Q4: I see that this has been included in line 483. I am glad the authors found the suggestion of the paper appropriate. If the authors feel that there is a better reference than Z Lin's I encourage them to use that instead. The reason I suggesting Z Lin's paper is it appeared to be the base for the ESM-2 technique under discussion.
> > Q5: I see the explanation has been added to Appendix D.1. This satisfies my concerns over this area.
> >
> > As a last note: I didn't mention this in the original review but I appreciate Appendix A. The explanation provided there makes it much more convenient to understand the techniques presented in the paper.
> >
> > Given these updates, I am happy to update the score from a 6 to an 8.

---

### Official Review · Reviewer_9Yfz · 2024-11-03

**Soundness:** 3
**Presentation:** 3
**Contribution:** 3
**Rating:** 8
**Confidence:** 3

**Summary:**

This paper introduces a new method for family-based protein generation, as a cross between mutation-based (where a single sequence is mutated one residue at a time) and de novo methods (where the design is entirely structure- or function-driven and the sequence is inferred from scratch). It does by adopting Bayesian Flow Networks. It operates in sequence space and is thus most analogous to a handful of recent methods on sequence-based diffusion, which is in general an underexplored space for protein design.

**Strengths:**

* The method is novel and explores a region of algorithm space that is generally underexplored for proteins, namely sequence-based design and in particular MSA-based design.
* Results are competitive if not necessarily state of the art. I consider the paper more about new ideas and approaches as opposed to necessarily achieving the best results.
* The method essentially finds a niche between quality and novelty of generated structures, sitting in between existing methods on the Pareto frontier.
* Experiments are well done and the algorithm itself appears clever.

**Weaknesses:**

* Biggest weakness (see question) is that how the evaluations are precisely done is not well described at all.

**Questions:**

* In Table 1, structural metrics are described (LR P@L) but how structures are predicted is actually never described. The paper repeatedly refers to CAMEO but CAMEO is not a method, it’s an evaluation set. Are actual structures being predicted using AlphaFold or similar method and then assessed? Or are only contacts being predicted using e.g., Potts models and then assessed? This is never stated in the main text or the appendix as far as I can tell.
* Similarly in Table 2, percentages are provided for what fractions of enzymes are considered functional, but the method of assessment is never mentioned! How is an enzyme being assessed as being functional? Is this based on a prediction tool? If so which tool?
* Similarly in Table 3, the focus is no longer on just enzymes but function prediction in general, but again how any of the accuracies are computed is never mentioned.

If these ambiguities are properly addressed I would increase my score.

---

> ### Author Response · Authors · 2024-11-20
> **Response to Reviewer 9Yfz (1/3)**
>
> We appreciate the reviewer's constructive comment and would like to address the questions as follows:
>
> **Q1: In Table 1, structural metrics are described (LR P@L) but how structures are predicted is actually never described. The paper repeatedly refers to CAMEO but CAMEO is not a method, it’s an evaluation set. Are actual structures being predicted using AlphaFold or similar method and then assessed? Or are only contacts being predicted using e.g., Potts models and then assessed? This is never stated in the main text or the appendix as far as I can tell.**
>
> We appreciate the opportunity to clarify our approach to structural metric collection and prediction.
>
> - Regarding our use of the term "CAMEO", we acknowledge the prior ambiguity and appreciate your feedback. CAMEO is indeed an evaluation dataset rather than a method, and we have corrected this in the main text to consistently refer to it as a dataset collected from CAMEO. These revisions ensure clarity and correctness in our description.
>
> - Concerning the structural metrics in Table 1, we apologize for any confusion. The contacts are predicted using a Potts model implemented in CCMPred. These predicted contact maps are then evaluated against the ground truth contact maps from the CAMEO dataset.
>
> - The metrics **LR P@L**, **LR P@L/2**, and LR **P@L/5** represent the precision at L, L/2, and L/5, respectively. Given predicted contact confidence score, the calculation of LR P@L/K is as follows:
> > **Filter Long-Range Contacts**: From the list of all predicted contacts, select only those that are long-range ($|i - j| \geq 24$ where $i$ and $j$ are sequence positions).
> >
> > **Sort by Confidence**: Sort the predicted long-range contacts based on their confidence scores.
> >
> > **Select Top L/K**: Choose the top L/K contacts from the sorted list.
> >
> > **Identify True Positives**: Determine which of these top L/K contacts are true contacts by comparing them to ground truth contact. Denote the number of true positives as $\text{TP}\_{\text{L/K}}$.
> >
> > The resulting precision at L/K is then calculated as:
> > $$
> > \text{LR @L/K} = \frac{\text{TP}\_{\text{L/K}}}{L/K}
> > $$
>
> - The **Diversity** metric **Div.** in Table 1 is calculated as the average pairwise sequence identity among the generated sequences. A lower diversity score indicates a more diverse set of sequences.
> - The **Novelty** metric **Nov.** in Table 1 is calculated as the maximum identity between the generated sequences and natural sequences, defined as $\frac{\sum_i(1 - \underset{j}{\max}(\text{identity}\_{ij}))}{N}$, where  $\text{identity}\_{ij}$  denotes the identity between $i$th among $N$ generated sequence and $j$th reference sequence.
>
> Necessary change have been made to the revised paper
>
> **Q2: Similarly in Table 2, percentages are provided for what fractions of enzymes are considered functional, but the method of assessment is never mentioned! How is an enzyme being assessed as being functional? Is this based on a prediction tool? If so which tool?**
>
> We appreciate the opportunity to elaborate on our methodology for assessing enzyme functionality.
>
> In Table 2, we classify and evaluate the generated enzymes using the enzyme function prediction model, CLEAN [1]. Our focus is on three representative categories of catalytic enzymes, each with extensive experimental validation. The models generate new enzymes based on reference sequences from these categories.
>
> We then employ CLEAN to predict the EC numbers of these newly generated enzymes, which allows us to assess their catalytic activity. A sequence is embeded by CLEAN model and compared with EC number's representation to obtain Euclidean distance to form a score, maximum separation were used to prioritize confident predictions of EC numbers from the ranking order.
>
> We have revised the main text to clarify this point (line 365-372), with changes clearly highlighted for reference. Furthermore, we've revised the paper to include more comprehensive results in Table 6, enhancing the clarity and depth of our findings.
>
> [1] Yu, Tianhao and Cui, Haiyang and Li, Jianan Canal and Luo, Yunan and Jiang, Guangde and Zhao, Huimin. "Enzyme function prediction using contrastive learning"

---

> ### Author Response · Authors · 2024-11-20
> **Response to Reviewer 9Yfz (2/3)**
>
> **Q3: Similarly in Table 3, the focus is no longer on just enzymes but function prediction in general, but again how any of the accuracies are computed is never mentioned.**
>
> Apologies for any confusion caused.  Table 3 outlines 2 task categories: regression and classification,  where protein thermostability prediction is a regresstion task, and the rest are classification tasks, the reason for the Fmax metric used in EC and GO task is due to the severe class inbalance problem.
>
> For all tasks, we follow the DPLM [1] and Saprot [2] methodologies by extracting the last hidden layer of our network to form a representation. This is then concatenated with a two-layer Multi-Layer Perceptron (MLP), configured according to DPLM and fine-tuned on each dataset. The MLP is optimized using the Mean Squared Error (MSE) loss for regression tasks and the cross-entropy loss for both binary and multi-class classification tasks.
>
> We now introduce each task in detail:
> | Task                              | Type                        | Training Set | Validation Set | Test Set | Description                                                                                                                                                        |
> |-----------------------------------|-----------------------------|--------------|----------------|---------|--------------------------------------------------------------------------------------------------------------------------------------------------------------------|
> | Protein Thermostability Prediction | Regression                  | 5,060        | 639            | 1,336   | Based on dataset used in FLIP[3]. Each protein receives a score for thermostability, evaluated with Spearman's correlation to actual values.                                |
> | HumanPPI Prediction                | Binary Classification       | 26,319       | 234            | 180     | Based on PEER[4], predicts interaction between two human proteins (Positive) or not (Negative).                                                                    |
> | Metal Ion Binding Prediction       | Binary Classification       | 4,247        | 1,066          | 1,083   | Determines presence of metal ion-binding sites in proteins (Positive or Negative) using a dataset from [5].                                                        |
> | EC (Enzyme Commission) Number Prediction | Multi-Class Classification | 13,089       | 1,465          | 1,604   | Based on dataset used in DeepFRI[6], classifying enzymes into one of 585 functional categories, with only one ground truth class considered positive.                       |
> | GO (Gene Ontology) Annotation Prediction | Multi-Class Classification | 26,224       | 2,904          | 3,350   | Based on DeepFRI[6], includes MF (Molecular Function) 489 classes, BP (Biological Process) 1,943 classes, and CC (Cellular Component) 320 classes  annotations. Only one ground truth category is considered positive.       |

---

> ### Author Response · Authors · 2024-11-20
> **Response to Reviewer 9Yfz (3/3)**
>
> We introduce the evaluation metrics as follows:
>
> **Spearman's rank correlation** (Spearman's $\rho$) [8] coefficient is a statistical measure that evaluates the strength and direction of the association between two ranked variables. It quantifies the degree of monotonicity in the relationship. In essence, it indicates whether an increase in one variable consistently corresponds to an increase or decrease in the other.
> Specifically, it is calculated as follows,
>
> $ \rho = 1 - \frac{6 \sum d_i^2}{n(n^2 - 1)}, ~ d_i = \widehat{y}_i - y_i$
>
> the prediction and ground truth are both ranked in descending order where $\widehat{y}_i$ and $y_i$ indicate the predicted and ground truth rank.
>
> **Accuracy** refers to the percentage of instances where the model accurately predicts the correct class for specific proteins in general it is computed as $\frac{\sum_1^N{\mathbf{1}_{(y=\hat{y})}}}{N}$, where $y$, $\hat{y}$ are the ground truth label and model predicted label, $\mathbf{1}\_{(\cdot)} $ is a Kronecker delta function, $N$ is the total number of samples.
>
>
> **Fmax (Maximum F1-score)**  metric is often used to evaluate tasks where there is an imbalance in the distribution of classes. Fmax is particularly useful because it reflects the balance between precision and recall, two important measures of a model's performance.
>
> Given $N$ model predicted scores $\\{s_i \\in [0, 1] \\}\_{i=1}^{N}$, corresponding labels are $\\{l\_i \in [0, 1] \\}\_{i=1}^{N}$, the True Positive (TP), False Positive (FP), True Negative (TN), and False Negative (FN) Precision (P), Recall (R), F1 score (F1) and finally $\text{F}\_{\text{max}}$ are subsequently calculated as follows:
>
> $ N\_{TP}(\lambda) = \\sum_{i}{l\_i\mathbf{1}_{(s\_i \\ge \\lambda)}}$,
>
> $ N\_{FP}(\lambda) = \\sum_{i}{l\_i\mathbf{1}_{(s_i < \\lambda)}}$,
>
> $ N\_{TN}(\lambda) = \\sum\_{i}{(1-l\_i)\mathbf{1}\_{(s\_i < \\lambda)}} $,
>
> $ N\_{FN}(\lambda) = \\sum\_{i}{(1-l\_i)\mathbf{1}\_{(s\_i \\ge \\lambda)}}$,
>
>  where $\mathbf{1}\_{(\cdot)} $ is a Kronecker delta function.
>
> $ P(\lambda) = \\frac{N\_{TP}(\lambda)}{N\_{TP}(\lambda) + N\_{FP}(\lambda)} $,
>
> $ R(\lambda) = \\frac{N\_{TP}(\lambda)}{N\_{TP}(\lambda) + N\_{FN}(\lambda)} $.
>
> The F1-score is subsequently calculated as $ F1 (\\lambda) = \\frac{2P(\lambda)R(\lambda)}{P(\lambda) + R(\lambda)} $.
>
> $\text{F}\_{\text{max}}$ is calculated by $\text{F}\_{\text{max}} = \\underset{\\lambda}{\\max} (F1(\\lambda))$,
>
> Details are included in the revised paper in Appendix D.2.3, and main text have also been revised and made necessary reference to the evaluation metric (line 390-398).
>
> [1] Wang, Xinyou and Zheng, Zaixiang and Ye, Fei and Xue, Dongyu and Huang, Shujian and Gu, Quanquan. "Diffusion Language Models Are Versatile Protein Learners"
>
> [2] Su, Jin and Han, Chenchen and Zhou, Yuyang and Shan, Junjie and Zhou, Xibin and Yuan, Fajie. "Saprot: Protein language modeling with structure-aware vocabulary"
>
> [3] Christian Dallago, Jody Mou, Kadina E. Johnston, Bruce J. Wittmann, Nicholas Bhattacharya, Samuel Goldman, Ali Madani, and Kevin K. Yang. "Flip: Benchmark tasks in fitness land-scape inference for proteins"
>
> [4] Minghao Xu, Zuobai Zhang, Jiarui Lu, Zhaocheng Zhu, Yangtian Zhang, Chang Ma, Runcheng Liu, and Jian Tang. "Peer: A comprehensive and multi-task benchmark for protein sequence understanding"
>
> [5] Mingyang Hu, Fajie Yuan, Kevin Yang, Fusong Ju, Jin Su, Hui Wang, Fei Yang, and Qiuyang Ding. "Exploring evolution-aware & -free protein language models as protein function predictors"
>
> [6] Vladimir Gligorijević, P. Douglas Renfrew, Tomasz Kosciolek, Julia Koehler Leman, Daniel Berenberg, Tommi Vatanen, Chris Chandler, Bryn C. Taylor, Ian M. Fisk, Hera Vlamakis, Ramnik J. Xavier, Rob Knight, Kyunghyun Cho, and Richard Bonneau. "Structure-based protein function prediction using graph convolutional networks"
>
> [7] José Juan Almagro Armenteros, Casper Kaae Sønderby, Søren Kaae Sønderby, Henrik Nielsen, and Ole Winther. "DeepLoc: prediction of protein subcellular localization using deep learning"
>
> [8] Zar, Jerrold H. "Spearman rank correlation"

---

> > ### Comment · Reviewer_9Yfz · 2024-11-25
> >
> > Thanks for addressing my concerns and improving the explanations of your assessments. I have increased my score to 8.

---

### Official Review · Reviewer_xv7N · 2024-11-03

**Soundness:** 2
**Presentation:** 2
**Contribution:** 2
**Rating:** 6
**Confidence:** 4

**Summary:**

The authors build a generative model of protein sequences based on the Bayesian Flow Network model, which they adapt to categorical data. They show that, conditioned on a protein family, their model better predicts structure and generates more realistic proteins than previous models as measured by activity and structure predictors.

**Strengths:**

They use a new architecture to model protein data.
They have many different sources of validation which have the potential to be convincing.

**Weaknesses:**

I strongly suggest the authors write their introduction more carefully. 1) de novo design, even if it has a low success rate, can be useful for library design. It is not necessarily to be contrasted with directed evolution as an alternative; in practice, libraries designed with these methods are used to find a hit which is then optimized by directed evolution. 2) The statement "However, it tends to produce proteins too similar to the original wild type, limiting the exploration of protein space and hindering the discovery of globally optimal proteins with desired functions." seems intuitively sound but is not backed up by empirical data in the literature. Given these comments as well, it makes little sense to position their "protein family design" as a third way (this paragraph should also compare their method with PoET and EvoDiff); I would appreciate them putting it in context in a library -> optimization pipeline. The related works section has a similar issue.

The paper understates the methods that previously have modeled protein families. I would like the authors to make it very clear that PoET, and ProtMamba (https://www.biorxiv.org/content/10.1101/2024.05.24.595730v1, which was not cited), did something like this previously; the authors mention some of these works in the experimental section, but they should be clearly mentioned in the introduction and related works section. I appreciate that the authors take an alternative approach to these models (which train on protein families); I would therefore also like an argument for why the authors expect their method to outperform these methods in principle given that one might expect their method to perform worse since they didn't train on protein family data.

The introduction of their model is too colloquial. It would strengthen the paper for the authors to introduce BFNs by describing the generative process and likelihood normally ("VLB stands for variational lower bound" for example). This lack of rigour is manifest in many of the ways the authors use the model (see questions).

"Consequently, we can bypass the need to construct an MSA profile training set, offering a more efficient and practical solution." This solution is more "efficient and practical" for the authors not the practitioners or those who wish to use the model; the authors should simply curate a training dataset as other model developers have done many times.

Important aspects of all of the evaluations are in the appendix, including the definition of nearly every metric. Any detail needed to understand the soundness of the evaluation (how you predict the enzymatic activity) should be in the main text.


In terms of soundness, conditioned on a protein family, one wishes to sample more sequences from that family. "diversity" and "novelty" are not the goal -- these statistics should be compared to those of the real family.

**Questions:**

What does the sampling process have to do with the generative process described by a BFN? It seems you're "denoising" a profile, but your claim is that the generative model itself outputs profiles. Please be as rigorous in your answer as possible.

In "Sampling Process Reflects Protein Conservation" the authors show a correlation between conservation at a site and transition rate. Is this something we expect? Is this something we should be interested in?

In Table 3 the authors show that the representations of their model can be used to predict downstream tasks (by the way, it is misleading to bold their results when they do not beat the baselines). Conspicuously missing is mutation effect prediction on ProteinGym. Is there a reason your model was not benchmarked on this when arguing for "understanding"?

Could you release representative sequences suggested by all models for table 2? Were their diversity or similarity to the conditioned family significantly different? How mush of the protein family did you pass to PoET to do this evaluation -- could you be clearer on how the baselines were run?

---

> ### Author Response · Authors · 2024-11-20
> **Response to Reviewer xv7N (1/5)**
>
> Thank you for your thoughtful feedback and suggestions, they will be invaluable in helping us improve our work. We address the raised concerns as follows:
>
> **W1: Towards the representation of Introduction:**
> - Statement over de novo design:
> We thank the reviewer for the comprehensive suggestions. We agree that de novo design and directed evolution are not necessarily contrasting methods but are often used in conjunction to enhance protein engineering efforts as you mentioned. In light of this, we have carefully revised the introduction to reflect the complementary relationship between de novo design and directed evolution, which can be found in line 35-43 of the introduction. Specifically, we include more discussions over the combination of directed evolution and de novo design, also involving a brief review of the most related works over the corresponding direction.
>
> - Towards the statement "However, it tends to produce proteins too similar to the original wild type, limiting the exploration of protein space and hindering the discovery of globally optimal proteins with desired functions." :
>
> Thank you for your insightful comment.  We agree that the statement could be better represented for strictness. After carefully checking the relevant literature, we revised the statement as follows:
>
> > Directed evolution[1,2] is effective in developing proteins with enhanced functions in vitro. However, the scope of exploration within the vast protein sequence space remains limited due to constraints in both the throughput of library creation and the subsequent screening or selection processes[3,4].
>
> The statement has been revised to reflect this explanation and make the statement more strict.
>
> **W2: Towards the position over “protein family design” and comprehensiveness of the Related Work section.**
>
> We agree that you suggested library -> optimization pipeline is more generalized for organizing the different directions in the field. To this end, we revised the corresponding part in the main text where "protein family design" could be understood as an intermediate approach that trades off both the library extension and sequence optimization. For a more detailed discussion refer to Line 44 to Line 52 in the revised draft.
>
> Sorry for missing the discussion of the previous protein family modeling tasks, we have revised the related work section (line 508-515) to make it more comprehensive. Specifically, we include the comparisons of the proposed approach with PoET, ProtMamba, MSAGPT, and EvoDiff, as recommended by the reviewer.
>
> [1] FRANCES H. ARNOLD. "Design by Directed Evolution"
>
> [2] Michael S. Packer and David R. Liu. "Methods for the directed evolution of proteins"
>
> [3] Wang, Yajie and Xue, Pu and Cao, Mingfeng and Yu, Tianhao and Lane, Stephan T and Zhao, Huimin. "Directed evolution: methodologies and applications"
>
> [4] Jesse D. Bloom and Frances H. Arnold. "In the light of directed evolution: Pathways of adaptive protein evolution"

---

> ### Author Response · Authors · 2024-11-20
> **Response to Reviewer xv7N (2/5)**
>
> **W3: The introduction of their model is too colloquial. It would strengthen the paper for the authors to introduce BFNs by describing the generative process and likelihood normally ("VLB stands for variational lower bound" for example).**
>
> Thank you for the constructive suggestions. As suggested, we have revised the introduction of the model to provide a more formal and detailed description of the model, and we include extra discussions of  VLB and the generative process in the following. More details could refer to Line 113-118 and line 228-240 in the revised draft.
>
> VLB discussion in Section 2.2 has been updated as follows:
> > In the context of a bits-back coding transmission scheme, the total number of nats required to transmit $\mathbf{x}$ with $\mathbf{z}\_{1:n}$ serving as intermediate latents can be expressed as $-\log p(\mathbf{z}\_{1:n}) - \log p(\mathbf{x}|\mathbf{z}\_{1:n})$. The process also incorporates $-\log q(\mathbf{z}\_{1:n}| \mathbf{x})$ nats returned to the sender, thus yielding the expected marginal nats necessary to transmit data from $p(\mathbf{x})$, which corresponds to the negative Variational Lower Bound (VLB)
>
> Generative process in Section 3.3 has been updated as follows:
> > Given a protein family profile $\left(\mathbf{P}^{(i)}\right)\_{i=1}^m \subset \Delta^{K-1}$, we first compute it's Bayesian flow up to some initial time step $t_0$, then
> >for $j$ in $\left[0, \cdots, N\right], ~ t_j \leftarrow \frac{(1-t_0)j}{N} + t_0$  do the following calculation iteratively:
> >$$
> >\mathbf{\theta}\_{t_{j}}^{(i)} \sim p\_F(\mathbf{\theta} | \mathbf{P}\_{\mathbf{\phi};j}^{(i)};t\_j), \\
> >$$
> >$$
> >\mathbf{P}\_{\mathbf{\phi};{(j+1)}} = f\_\mathbf{\phi}(\mathbf{\theta}\_{t\_{j}}^{(1)}, \cdots, \mathbf{\theta}\_{t\_{j}}^{(m)}, t\_j),
> >$$
> >Where the initial $\left(\mathbf{P}\_{\mathbf{\phi};0}^{(i)}\right)\_{i=1}^m$ is set to $\left(\mathbf{P}^{(i)}\right)\_{i=1}^m$.
> >Finally we take the $\text{argmax}$ sampling over $\left(\mathbf{P}\_{\mathbf{\phi};(N+1)}^{(i)}\right)\_{i=1}^m$ to get the generated family protein sequence, the $i$th amino acid can be decoded as follows: $a^{(i)} = \text{argmax}\_k (\mathbf{P}_{\mathbf{\phi};(N+1)}^{(i)})\_{k}$.
>
> Moreover, we have added Algorithm 1 (line 989-1002) for the Training Loss Procedure and Algorithm 2 (line 1004-1018)for the Family Protein Sampling Procedure, to provide a more formal and detailed description of the model.
>
> **W4: The argument over why the proposed approach outperforms other methods in principle.**
>
> Thanks a lot for your insightful comments which also inspire us to further explore the potential of the proposed approaches.
>
> Firstly, to provide a more strict statement, we fine-tuned our model with MSA profiles from Uniref50 clusters for empirical support. The results can be found in the following **Table 1** of response to **W5** where the performance of the proposed approach benefits from the protein family data.
>
> Secondly, we propose several possible explanations of why our sequence-training model could outperform MSA-training methods:
>
> - Statistically, proteins within the same family often have similar sequences. From a generative model's compression perspective, the model tends to encourage similar data points to share a common representation. This implies that the model might inherently capture family information from just a single sequence.
>
> - Compared to autoregressive models such as MSAGPT, PoET, and ProtMamba, non-autoregressive modeling is more suitable for proteins. This is because protein structures have spatial constraints requiring flexible dependencies, rather than a strict left-to-right dependence.
>
> - From a computational efficiency standpoint, while autoregressive models are effective, they face challenges with long sequences, such as when concatenating a list of proteins from a family. For EvoDiff, the computational demand increases rapidly with the depth of the MSA, hindering scalable model training and resulting in small-scale models. More specifically, PoET contains 201M parameters, whereas EvoDiff has only 100M parameters.

---

> ### Author Response · Authors · 2024-11-20
> **Response to Reviewer xv7N (3/5)**
>
> **W5 the authors should simply curate a training dataset with MSA profiles as other model developers have done many times.**
>
> - This is a great suggestion. As mentioned in **W4**, we acknowledge that protein family data could potentially provide more co-occurrence information that would enhance the model and that single-sequence training can also implicitly capture family information.
> - The key question is how much improvement can be achieved by training with profiles constructed from the MSA of protein families. Given the limited discussion period, we have conducted preliminary experiments by fine-tuning with protein family profiles, with the results presented in **Table 1**,  We also present the table in the rebuttal. The fine-tuned model is featured in the **last row**. The results indeed improve after tuning with MSA profiles. For a more comprehensive training approach, we plan to explore this direction deeply in future work.
>
> | Model                           | Sequence Div &darr; | Sequence Nov &uarr; | Structure LR P@L &uarr; | Structure LR P@L/2 &uarr; | Structure LR P@L/5 &uarr; |
> | ------------------------------- | ------------------- | ------------------- | ----------------------- | ------------------------- | ------------------------- |
> | Searched MSA                    | -                   | -                   | 0.186                   | 0.270                     | 0.395                     |
> | ESM-2 (150M)                    | 0.565               | **0.691**           | 0.086                   | 0.116                     | 0.167                     |
> | ESM-2 (650M)                    | 0.619               | 0.556               | 0.100                   | 0.146                     | 0.223                     |
> | PoET-Single (201M)              | 0.853               | 0.200               | 0.025                   | 0.028                     | 0.031                     |
> | PoET-MSA (201M)                 | 0.651               | 0.243               | 0.036                   | 0.042                     | 0.051                     |
> | EvoDiff-MSA (100M)              | **0.225**           | 0.668               | 0.061                   | 0.089                     | 0.168                     |
> | DPLM (150M)                     | 0.369               | 0.463               | 0.093                   | 0.147                     | 0.284                     |
> | DPLM (650M)                     | 0.445               | 0.411               | 0.102                   | 0.159                     | 0.303                     |
> | **profileBFN-Single (150M)**    | 0.368               | 0.646               | 0.126                   | 0.197                     | 0.321                     |
> | **profileBFN-Single (650M)**    | 0.421               | 0.581               | 0.162                   | 0.262                     | 0.422                     |
> | **profileBFN-Profile (150M)**   | 0.283               | 0.650               | 0.128                   | 0.210                     | 0.384                     |
> | **profileBFN-Profile (650M)**   | 0.293               | 0.641               | 0.173                   | 0.280                     | 0.474                     |
> | **profileBFN-Profile\* (650M)** | 0.284               | 0.653               | **0.176**               | **0.291**                 | **0.486**                 |
>
> **W6: Important aspects of all of the evaluations are in the appendix, including the definition of nearly every metric. Any detail needed to understand the soundness of the evaluation (how you predict the enzymatic activity) should be in the main text.**
>
> We thank the reviewer for the constructive criticism. We have revised the manuscript to include more detailed definitions and descriptions of the evaluation metrics in the main text. Additionally, we have ensured that all essential information needed to understand the evaluation's soundness, including how enzymatic activity is predicted, is now clearly presented.

---

> ### Author Response · Authors · 2024-11-20
> **Response to Reviewer xv7N (4/5)**
>
> **W7: In terms of soundness, conditioned on a protein family, one wishes to sample more sequences from that family. "diversity" and "novelty" are not the goal -- these statistics should be compared to those of the real family.**
>
> Thank you for your feedback. We agree with your point that "diversity" and "novelty" are not the primary goals of our protein sequence generation model. Our primary objective is to ensure soundness by generating sequences from the given protein family.
>
> However, we believe that a comprehensive evaluation—including metrics of diversity and novelty—is still necessary to assess the model's performance and performance effectively. While these metrics are not our main focus, they provide important insights into how well our model captures the variation inherent in the protein family without collapsing into a single sequence or trivially copying from the input. We will add more discussions as you commented in the revised version for help the readers distinguish  the focus of different metrics.
>
> **Q1: What does the sampling process have to do with the generative process described by a BFN? It seems you're "denoising" a profile, but your claim is that the generative model itself outputs profiles. Please be as rigorous in your answer as possible.**
>
> Apologies for any confusion.  We start with the definition of the unified profile, which incorporates both the protein sequences as the so-called single sequence profile and the real MSA profiles from the protein families.  The motivation for proposing a unified profile view is to incorporate natural evolutionary information from the huge amount of single sequences, as well as expert human insights on co-evolutionary patterns derived from multiple sequence alignments (MSAs).
>
> Our model is actually a profile generative model as you anticipated, which models and outputs profiles. However, in many application and evaluation settings,  we actually need the protein sequences of the generated profiles.  Hence, we would sample the sequence from generated profiles with a simple process, e.g. greedy search.
>
> To clarify any ambiguity, we have revised the paper to include more details. Specifically, we have added Algorithm 1 (line 989-1002) for the Training Loss Procedure and Algorithm 2 (line 1004-1018) for the Family Protein Sampling Procedure. Additionally, Section 3.3 (line 223) has been updated accordingly. All revisions are highlighted in blue for easy identification.
>
> **Q2: In "Sampling Process Reflects Protein Conservation" the authors show a correlation between conservation at a site and transition rate. Is this something we expect? Is this something we should be interested in?**
>
> Thank you for your question. We examined the variation at each position over multiple sampling iterations to understand how our model behaves. Some positions quickly concentrate on few amino acids, resulting in lower overall variation. Since we are generating proteins that belong to a specific family, it is expected that a successful generation will exhibit similar characteristics, such as conservation and co-evolution, to those of the family of interest.
>
> Therefore, it is natural to observe that the generation process mirrors these characteristics. This correlation between conservation at a site and transition rate is something we anticipate and should be interested in, as it indicates that the generated sequences reflect the intrinsic properties of the protein family.
>
> **Q3: In Table 3 the authors show that the representations of their model can be used to predict downstream tasks (by the way, it is misleading to bold their results when they do not beat the baselines). Conspicuously missing is mutation effect prediction on ProteinGym. Is there a reason your model was not benchmarked on this when arguing for "understanding"?**
>
> Thank you for the constructive criticism.
> - In our revisions, we have corrected some mistakenly bolded numbers and provided clarification on the results that remain bolded. Specifically, the reference baselines, SaProt and MIF-ST, incorporate structural information. Additionally, we compare our model with the reproduced version of the DPLM results.
>
> - We have additionally conducted experiments on ProteinGym, and on ClinVar Dataset as well, the details are provided in Appendix E.5, please refer to Table 9 for the results.
>
> | Dataset        | ESM-2 | ESM-1b | Tranception | ESM-IF | MIF-ST | EVE   | SaProt    | ProfileBFN |
> | :------------- | ----- | ------ | ----------- | ------ | ------ | ----- | --------- | ---------- |
> | **ClinVar**    | 0.862 | 0.900  | 0.945       | 0.748  | 0.891  | 0.878 | **0.909** | 0.901*     |
> | **ProteinGym** | 0.475 | 0.440  | 0.413       | 0.409  | 0.474  | -     | **0.478** | 0.476*     |
>
> The results indicate that ProfileBFN outperforms most methods except for Saprot, which  incorporates 3D structural information. This demonstrates ProfileBFN's capability in zero-shot representation.

---

> ### Author Response · Authors · 2024-11-20
> **Response to Reviewer xv7N (5/5)**
>
> **Q4: Could you release representative sequences suggested by all models for table 2? Were their diversity or similarity to the conditioned family significantly different? How much of the protein family did you pass to PoET to do this evaluation -- could you be clearer on how the baselines were run?**
>
> - We have included three figures in the Appendix—Figure 11 for P40925, Figure 12 for Q7X7H9, and Figure 13 for Q15165—that showcase the generated protein sequences.
> - We have revised Section 4.1 to provide more detailed information on the evaluation setup. Specifically, we have clarified why the enzymes are selected for the experiment and the evaluation of the generated sequences. Please refer to line 259-398 for these changes, which are highlighted in blue.
> - The protein sequences for P40925, Q7X7H9, and Q15165 comprise 572, 443, and 15 sequences, respectively. This information is now clearly presented in Table 6 of the manuscript. Additionally, we have expanded the evaluation by incorporating metrics such as Accuracy, Uniqueness, Novelty, and Diversity to offer a more comprehensive comparison.
>
> For convenience, we have included the following table in the rebuttal, the last row is the append experiment that is fine-tuned with MSA profiles.
>
> |                              | model              | P40925 | Q7X7H9 | Q15165 |
> | ---------------------------- | ------------------ | ------ | ------ | ------ |
> | MSA Depth                    | -                  | 572    | 443    | 15     |
> |                              |                    |        |        |        |
> | Accuracy × Uniqueness &uarr; | PoET               | 3.00%  | 33.3%  | 0.05%  |
> | Accuracy × Uniqueness &uarr; | EvoDiff-MSA        | 27.93% | 88.69% | 1.39%  |
> | Accuracy × Uniqueness &uarr; | ProfileBFN-profile | 95.19% | 98.98% | 42.67% |
> |                              |                    |        |        |        |
> | Accuracy &uarr;              | PoET               | 98.04% | 99.93% | 100%   |
> | Accuracy &uarr;              | EvoDiff-MSA        | 27.93% | 88.69% | 1.39%  |
> | Accuracy &uarr;              | ProfileBFN-profile | 95.19% | 98.98% | 42.67% |
> |                              |                    |        |        |        |
> | Uniqueness &uarr;            | PoET               | 3.06%  | 33.32% | 0.05%  |
> | Uniqueness &uarr;            | EvoDiff-MSA        | 100%   | 100%   | 100%   |
> | Uniqueness &uarr;            | ProfileBFN-profile | 100%   | 100%   | 100%   |
> |                              |                    |        |        |        |
> | Novelty &uarr;               | PoET               | 0.036  | 0.366  | 0.068  |
> | Novelty &uarr;               | EvoDiff-MSA        | 0.728  | 0.596  | 0.497  |
> | Novelty &uarr;               | ProfileBFN-profile | 0.467  | 0.582  | 0.288  |
> |                              |                    |        |        |        |
> | Diversity &darr;             | PoET               | 0.499  | 0.645  | 0.990  |
> | Diversity &darr;             | EvoDiff-MSA        | 0.138  | 0.184  | 0.143  |
> | Diversity &darr;             | ProfileBFN-profile | 0.374  | 0.289  | 0.594  |

---

> > ### Comment · Reviewer_xv7N · 2024-11-21
> > **Response**
> >
> > The primary strength of this paper is further investigation of what I feel is an incredibly promising research direction -- modeling of entire protein families.
> >
> > However it is held back by a poor exposition of what the purpose of this model is, its relation to previous work. The other reviews extoll the novelty of this paper seemingly unaware of previous work that was not clearly discussed. The authors address this somewhat in their new introduction, and I like the following paragraph from their rebuttal:
> > >we propose several possible explanations of why our sequence-training model could outperform MSA-training methods:
> > Statistically, proteins within the same family often have similar sequences. From a generative model's compression perspective, the model tends to encourage similar data points to share a common representation. This implies that the model might inherently capture family information from just a single sequence.
> > Compared to autoregressive models such as MSAGPT, PoET, and ProtMamba, non-autoregressive modeling is more suitable for proteins. This is because protein structures have spatial constraints requiring flexible dependencies, rather than a strict left-to-right dependence.
> > From a computational efficiency standpoint, while autoregressive models are effective, they face challenges with long sequences, such as when concatenating a list of proteins from a family. For EvoDiff, the computational demand increases rapidly with the depth of the MSA, hindering scalable model training and resulting in small-scale models. More specifically, PoET contains 201M parameters, whereas EvoDiff has only 100M parameters.
> >
> > And I would appreciate the authors replacing their fourth paragraph in the intro / motivation with something along these lines.
> >
> > The other major weakness is the ability of this model to actually model protein families since the authors use an alternative training objective that trains their model only on individual proteins: the text clearly is written to suggest they learn entire families, not individual proteins and I was surprised when I came across the final objective. Bafflingly, the authors actually could train their model on entire families and in the rebuttals showed that this leads to a better model. This, I'm sure, will lead to confusion once the paper is published. However, I appreciate the model is now actually trained on families now.
> >
> > Finally, the ambiguous use case for this model is manifest in the confused validation the authors perform. The authors did an enormous amount of work to compare their model to every other for a large number of use cases; a focussed paper could help practitioners know when to use this model and saved the authors a lot of work. They added more validation in the rebuttal (note the bolding is wrong for ClinVar, where Tranception seems to win).
> >
> > Acknowledging these downsides which I think the authors understand and will keep in mind when presenting this work, the authors put a heroic effort into their rebuttal and I recommend an accept without conditions.

---

### Author Response · Authors · 2024-11-26
**General Response**

We sincerely thank the reviewers for their time and for supporting our work. We are grateful for the valuable feedback provided, which has contributed to enhancing the quality and rigor of our paper. The constructive comments and insightful suggestions have been instrumental in guiding our revisions and have provided us with a deeper understanding of the areas.

We are committed to maintaining high standards in our research and writing, and your feedback has been invaluable in this endeavor. As we prepare the final version of the paper, we will continue to bear in mind all the comments and suggestions provided, ensuring that we produce a comprehensive and well-substantiated final manuscript.

Once again, thank you for your thoughtful review and for contributing to the development of our work.

---

### Meta-Review · Area_Chair_nJ1q · 2024-12-15

**Metareview:**

This paper adopts and extends Bayesian Flow Network (BFN) to propose ProfileBFN that enables MSA profile-based protein family design. Reviewers acknowledge the novelty of the proposed approach and its potential advantages, demonstrated in good empirical performance and its potential to improve protein family design. Reviewers have noted some room for improvement in the overall presentation of the paper. Overall, this paper presents an interesting work with novel contributions that are worth sharing with the relevant research community.

**Additional Comments On Reviewer Discussion:**

The authors have actively responded to the reviewers' concerns and suggestions, providing additional evidence that further demonstrates the performance and advantages of the proposed ProfilBFN and clarifying initial ambiguities in the original manuscript.
Reviewers note that the authors' rebuttals have addressed most of their original concerns and increased their confidence regarding the merits of the proposed method.

---

### Decision · Program_Chairs · 2025-01-22

Accept (Oral)